

# Modeling oceanic nitrite concentrations and isotopes using a 3D inverse N cycle model

Taylor S. Martin[1], François Primeau[2], Karen L. Casciotti[1*]

[1]Stanford University, Department of Earth System Science
[2]University of California, Irvine, Department of Earth System Science

[*]Corresponding author: kcasciotti@stanford.edu; 650-721-5545

**Abstract.** Nitrite ($NO_2^-$) is a key intermediate in the marine nitrogen (N) cycle and a substrate in nitrification, which produces nitrate ($NO_3^-$), as well as water column N loss processes, denitrification and anammox. In models of the marine N cycle, $NO_2^-$ is often not considered as a separate state variable, since $NO_3^-$ occurs in much higher concentrations in the ocean. In oxygen
deficient zones (ODZs), however, $NO_2^-$ represents a substantial fraction of the bioavailable N, and modeling its production and consumption is important to understanding the N cycle processes occurring there, especially those where bioavailable N is lost from or retained within the water column. Here we present the expansion of a global 3D inverse N cycle model to include $NO_2^-$ as a reactive intermediate as well as the processes that produce and consume $NO_2^-$ in marine ODZs. $NO_2^-$ accumulation in ODZs is accurately represented by the model involving $NO_3^-$ reduction, $NO_2^-$ reduction, $NO_2^-$ oxidation, and anammox. We model both $^{14}N$ and $^{15}N$ and use a compilation of oceanographic measurements of $NO_3^-$ and $NO_2^-$ concentrations
and isotopes to place a better constraint on the N cycle processes occurring. The model is optimized using a range of isotope effects for denitrification and $NO_2^-$ oxidation, and we find that the larger (more negative) inverse isotope effects for $NO_2^-$ oxidation along with relatively high rates of $NO_2^-$ oxidation give a better simulation of $NO_3^-$ and $NO_2^-$ concentrations and isotopes in marine ODZs.

## 1 Introduction

Nitrogen (N) is an important nutrient to consider when assessing the biogeochemical cycling in the ocean. The N cycle is intrinsically tied to the carbon (C) cycle, whereby N can be the limiting nutrient for primary production and carbon dioxide uptake (Moore et al., 2004; Codispoti, 1989). Understanding the distribution of nitrogen in the ocean allows us to make
inferences about the effects on other nutrient cycles and potential roles that N may play in a regime of climate change (Gruber, 2008).

There are several chemical species in which N can be found in the ocean. The largest pool of bioavailable N is nitrate ($NO_3^-$), a dissolved inorganic species, which can be taken up by microbes for use in assimilatory or dissimilatory processes. Another



dissolved inorganic species, nitrite ($NO_2^-$), accumulates in much lower concentrations but is a key intermediate in many N cycling processes. Models of the marine N cycle often include $NO_3^-$ and $NO_2^-$ together as a single dissolved inorganic N (DIN) pool, or exclude $NO_2^-$ entirely (DeVries et al., 2013; Deutsch et al., 2007; Brandes and Devol, 2002). However, $NO_2^-$ does accumulate significantly in oxygen deficient zones (ODZs) in features known as secondary $NO_2^-$ maxima (SNMs), and it is an

intermediate or substrate in many important N cycle processes occurring there. ODZs are hotspots for marine N loss (Codispoti et al., 2001; Deutsch et al., 2007), which is driven by processes that result in conversion of bioavailable DIN to dinitrogen gas ($N_2$). The two main water column N loss processes, denitrification and anammox, use $NO_2^-$ as a substrate. Denitrification involves the stepwise reduction of $NO_3^-$ to $NO_2^-$ and then to gaseous products nitric oxide (NO), nitrous oxide ($N_2O$), and $N_2$. Anammox consists of the anaerobic oxidation of ammonium ($NH_4^+$) to $N_2$ using $NO_2^-$ as the electron acceptor. $NO_2^-$ is also

oxidized to $NO_3^-$ during anammox, representing an alternative fate for $NO_2^-$ in ODZs. Indeed, $NO_2^-$ oxidation appears to be prevalent in ODZs, with more $NO_2^-$ oxidation occurring than can be explained by anammox alone (Gaye et al., 2013; Peters et al., 2016; Peters et al., 2018; Babbin et al., 2017; Buchwald et al., 2015; Casciotti et al., 2013; Martin and Casciotti, 2017). $NO_2^-$ oxidation results in the regeneration of $NO_3^-$ that would otherwise be converted to $N_2$ and lost from the system. The close coupling between $NO_3^-$ reduction to $NO_2^-$ and $NO_2^-$ oxidation back to $NO_3^-$ represents a kind of control valve on the marine N

budget (Penn et al., 2016; Bristow et al., 2016). Where $NO_2^-$ oxidation can outcompete $NO_2^-$ reduction via denitrification and anammox, bioavailable N is retained. Where $NO_2^-$ oxidation is limited by oxygen availability, then N loss can occur. Thus, understanding the $NO_2^-$ dynamics in ODZ waters is critical to assessing the N loss occurring there.

The observed $NO_3^-$ and $NO_2^-$ concentrations alone do not allow us to fully characterize the N cycling processes occurring in a

given region. Stable isotope measurements of $NO_3^-$ and $NO_2^-$ provide additional insight and constraints on marine N cycle processes. There are two stable isotopes of N, $^{14}N$ and $^{15}N$. The isotopic ratios for a given N species, usually expressed in delta notation as $\delta^{15}N$ (‰) = ($^{15}N/^{14}N_{sample} \div {}^{15}N/^{14}N_{standard} -1) \times 1000$, is an integrated measure of the processes that have produced and consumed that N species. Each process imparts a unique isotope effect ($\varepsilon = {}^{14}k/^{15}k -1 \times 1000$, where $^{14}k$ and $^{15}k$ are the first order rate constants for the $^{14}N$ and $^{15}N$ containing molecules, respectively) that impacts the isotopic composition of the

substrate and the product (Mariotti et al., 1980). In particular, $NO_2^-$ cycling processes have distinct isotope effects for $NO_2^-$ reduction, which occurs with normal isotopic fractionation (Bryan et al., 1983; Martin and Casciotti, 2016; Brunner et al., 2013) and $NO_2^-$ oxidation, which occurs with an unusual inverse kinetic isotope effect (Casciotti, 2009; Buchwald and Casciotti, 2010; Brunner et al., 2013). Thus, the isotopes of $NO_2^-$ are very sensitive to the relative importance of $NO_2^-$ oxidation and $NO_2^-$ reduction in $NO_2^-$ consumption (Casciotti, 2009; Casciotti et al., 2013).

Models of the marine N cycle have employed isotopes and isotope effects in conjunction with N concentrations to elucidate N cycle processes (Brandes and Devol, 2002; Sigman et al., 2009; Somes et al., 2010; DeVries et al., 2013; Casciotti et al., 2013; Buchwald et al., 2015; Peters et al., 2016). A model can either assume a set of processes and infer the underlying isotope effects, or assume isotope effects and infer a set of processes. These isotope models are highly dependent on the chosen isotope



effects used for given processes. Though there are estimates of isotope effects for processes based on both environmental measurements and in-lab studies, there is not always agreement between them. For example, laboratory cultures of $NO_2^-$ oxidizers indicate an N isotope effect of $^{15}\varepsilon$ = -13‰ (Casciotti, 2009), while measured concentrations and isotopes of $NO_3^-$ and $NO_2^-$ in ODZs indicate that an isotope effect closer to -30‰ is more consistent with the observations (Buchwald et al.,

2015; Casciotti et al., 2013).

Here we present an expansion of an existing global ocean 3D inverse isotope-resolving N cycling model (DeVries et al., 2013) to include $NO_2^-$ and its isotopes as tracers. The addition of $NO_2^-$ allows us to include many additional internal N cycling processes, as well as a more nuanced and realistic version of the processes occurring in ODZs. This should result in a

reasonably accurate global representation of $NO_3^-$ and $NO_2^-$ concentrations and isotopes. We used a database of $NO_3^-$ and $NO_2^-$ observations in order to assess the performance of the model as well as optimize the model N cycle parameters for which we do not have good prior estimates. In the model we employ a variety of isotope effect estimates for three important ODZ processes—$NO_3^-$ reduction, $NO_2^-$ reduction, and $NO_2^-$ oxidation—to discern what isotope effect estimates result in the best fit to the observations.

**2 Methods**

**2.1 Inverse nitrogen cycle model overview**

The model used here is a steady-state inverse model that solves for the concentration and $\delta^{15}N$ of $NO_3^-$, $NO_2^-$, particulate organic N (PON), and dissolved organic N (DON) using a set of linear equations. Because the model assumes that the system is in steady state, it is not able to capture time dependent properties of the system such as seasonality. However, on inter-annual

timescales the N cycle is thought to be approximately in balance (Gruber, 2004; Bianchi et al., 2012). The residence time of N in the ocean, which is thought to be on the order of 2000-3000 years (Gruber, 2008), is sufficiently long to preclude any detectable changes in the global N inventory to date on timescales commensurate with the global overturning circulation. An important advantage of the steady-state assumption for our linear model is that it is possible to find solutions by direct matrix inversion without the need for a spin-up period as is required by forward models. The solution to the system provides $^{14}N$ and

$^{15}N$ concentrations of the N species of interest at every grid point in the model system. Working with a linear system imposes some restrictions on how complicated the rate equations can be, but there are improvements in model performance and ease of use, allowing us to test hypotheses about the processes that govern the marine N cycle and budget. We aimed to produce a realistic N cycle model that represented ODZ processes accurately while limiting the number of free parameters. The description below outlines the dependencies and simplifications employed in this version of the model.

The model's uncertain biological parameters were determined through an optimization process that minimizes the difference between the modeled and observed $NO_3^-$ and $NO_2^-$ concentration and isotope data. Computational time limits the number of





parameters that we were able to optimize. We therefore focus our investigation on parameters that are poorly constrained by literature values and to which the model solution is most sensitive. In order to determine the parameters for optimization, a sensitivity analysis was performed on each parameter, varying them individually by ±10% and computing the change in the modeled $^{14}$N and $^{15}$N. Those exhibiting variability of >5% were chosen for optimization in the model. The sensitivity analysis

and the optimal values of the parameters contribute to an improved understanding of the cycling of nitrogen in the ocean in general and in the ODZs in particular.

The sensitivity analysis revealed that the modeled distribution of $^{15}$N was very sensitive to isotope effects, parameters that control the relative rates of $^{15}$N and $^{14}$N in chemical and biological processes. There are literature estimates each of the isotope

effects of interest in this work, although there is often a discrepancy between isotope effects estimated in laboratory studies and those expressed in oceanographic measurements (Casciotti et al., 2013; Buchwald et al., 2015; Bourbonnais et al., 2015; Martin and Casciotti, 2017; Fuchsman et al., 2017; Peters et al., 2018b). Rather than optimizing the isotope effect values, we have chosen to use multiple cases with different combinations of previously estimated isotope effects in order to assess which values best fit the observations.

In addition to the optimized parameters and isotope effects, there were some non-sensitive parameters that were fixed prior to the optimization and whose values were chosen using literature estimates (Table 1). Some N cycle processes are also dependent on prescribed input fields that are not explicitly modeled, such as temperature, phosphate, and net primary production. These external input fields will be discussed in detail in the relevant sections for each N cycle process.

**2.2 Model grid and transport**

The model uses a uniform 2˚x2˚ grid with 24 depth levels. The thickness of each model layer increases with depth, from 36 m at the top of the water column to 633 m near the bottom. Bottom topography was determined using 2-minute gridded bathymetry (ETOPO2v2) that was then interpolated to the model grid. Our linear N cycle model relies on the transport of dissolved N species ($NO_3^-$, $NO_2^-$, and DON) in the ocean. For this we use the annual averaged circulation as captured by a

tracer transport operator that governs the rate of transport of dissolved N species ($NO_3^-$, $NO_2^-$, and DON) between boxes. The original version of the tracer data-assimilation procedure used to generate the transport operator for dissolved species ($T_f$) is described by DeVries and Primeau (2011), and the higher resolution version used here is described by DeVries et al. (2013).

**2.3 N cycle**

In the N cycling portion of the model, we track four different N species. There are two organic N (ON) pools: DON and PON.

There are also two dissolved inorganic N (DIN) pools: $NO_3^-$ and $NO_2^-$. We excluded ammonium ($NH_4^+$) from the model because it typically only accumulates in low concentrations throughout the ocean, and scarcity of data (especially $^{15}$N data) would make model validation difficult. A schematic diagram of the N cycle in the model is shown in Figure 1.





Because we are interested in both $^{14}$N and $^{15}$N concentrations of each N species to constrain the rate parameters, two sets of governing equations were employed: one that depends on $^{14}$N and another that depends on $^{15}$N. Generally, the rate for $^{15}$N processes was dependent on the rate of $^{14}$N processes and an isotopic fractionation factor α that is specific to each process and

5 substrate. By solving for steady-state solutions to both $^{14}$N and $^{15}$N concentrations, we were able to model global distributions of [NO$_3^-$], [NO$_2^-$], and their corresponding δ$^{15}$N values.

**2.3.1 N cycle parameterization**

We will first address the $^{14}$N equations and the general format of the N cycle in the model. Each equation is then broken down into its component parts for further explanation of the biological processes and their parameterization. The $^{15}$N equations and

10 isotope implementation will be discussed in a later section.

The governing equations for the $^{14}$N-containing ON and DIN state variables can be written as follows:

1. $\left[\frac{\partial}{\partial t} + T_f\right]^{14}NO_3^- = J_{14}^{dep} - J_{14}^{assim,NO3} - J_{14}^{NAR} + J_{14}^{NXR} + 0.3J_{14}^{AMX} - J_{14}^{sed}$

2. $\left[\frac{\partial}{\partial t} + T_f\right]^{14}NO_2^- = J_{14}^{AMO} - J_{14}^{assim,NO2} + J_{14}^{NAR} - J_{14}^{NXR} - J_{14}^{NIR} - 1.3J_{14}^{AMX}$

3. $\left[\frac{\partial}{\partial t} + T_f\right] DO^{14}N = \sigma(J_{14}^{fix} + J_{14}^{assim,WOA}) + J_{14}^{sol} - J_{14}^{remin}$

4. $\left[\frac{\partial}{\partial t} + T_p\right] PO^{14}N = (1 - \sigma)(J_{14}^{fix} + J_{14}^{assim,WOA}) - J_{14}^{sol}$

The model is assumed to be in steady state, thus the $\frac{\partial}{\partial t}$ term is 0. The J terms represent the source and sink processes for each state variable, expressed in units of mmol/m$^3$/yr and will be described in more detail below. Briefly, $J_{14}^{dep}$ is the spatially-

20 variable deposition of NO$_3^-$ from the atmosphere to the sea surface. $J_{14}^{assim,NO3}$ and $J_{14}^{assim,NO2}$ represent the assimilation of NO$_3^-$ and NO$_2^-$, respectively, by phytoplankton in the upper two box levels. This assimilated NO$_3^-$ produces DON and PON, with proportions set by a spatially-variable σ term. Assimilation in the DON and PON equations is represented by $J_{14}^{assim,WOA}$ and is dependent on WOA [NO$_3^-$] as described below. N$_2$ fixation is split between DON and PON with the same σ term. NO$_3^-$ reduction ($J_{14}^{NAR}$), NO$_2^-$ reduction ($J_{14}^{NIR}$), NO$_2^-$ oxidation ($J_{14}^{NXR}$), and anammox ($J_{14}^{AMX}$) act on the NO$_3^-$ and NO$_2^-$ pools. $J_{14}^{sed}$

represents the removal of NO$_3^-$ via benthic denitrification. $J_{14}^{sol}$ represents the dissolution of PON into DON. $J_{14}^{remin}$ is related to ammonia oxidation ($J_{14}^{AMO}$) and $J_{14}^{AMX}$ as described below.

Through the use of these J terms, the governing equations are all linear with respect to the state variables. In order to introduce dependence of rates on the concentrations of multiple state variables, we run the ON equations first and the DIN equations

second. When [DON] is found in the [DIN] governing equations, that [DON] value has already been determined for each grid



box from the ON model. When [NO₃⁻] is found in the DON governing equations, it is drawn from World Ocean Atlas (WOA13) annual data interpolated to the model grid.

**2.3.2 N source processes**

**Atmospheric deposition**

Deposition of N to the ocean from the atmosphere is one of two sources of bioavailable N supplied to the ocean in this model. N deposition is assumed to only occur in the top box of the model, and we assume that most of the N deposited is as NO₃⁻, and that the other species would be rapidly oxidized to NO₃⁻ in the oxic surface waters.

5.   $J_{14}^{dep} = r_{14}^{dep} S^{dep}$

To calculate $J_{14}^{dep}$, the atmospheric deposition rate of $^{14}N$, we use modeled total inorganic N deposition for 1993, $S^{dep}$

(Galloway et al., 2004; Dentener et al., 2006; data available online at https://daac.ornl.gov/CLIMATE/guides/global_N_deposition_maps.html), which was interpolated to our model grid. This term, $S^{dep}$, is then multiplied by a prescribed fractional abundance of $^{14}N$ in the deposited N ($r_{14}^{dep}$), which is calculated from the isotopic composition of deposited N ($\delta^{15}N_{dep}$, -4‰; Equation 6), to yield the deposition of $^{14}N$ to the sea surface in each box ($J_{14}^{dep}$). To calculate $r_{14}^{dep}$ from $\delta^{15}N_{dep}$, we first calculate $r_{15}^{dep}$ using $r_{15}^{air}$, a standard with a value of 0.003676 (equation 6).

6.   $r_{15}^{dep} = \left(\frac{\delta^{15}N_{dep}}{1000} + 1\right) \times r_{15}^{air}$

Then, assuming that $[^{14}N] \sim [^{14}N] + [^{15}N]$, we calculate $r_{14}^{dep}$ as $(1 - r_{15}^{dep})$. The units of $S^{dep}$ are given in mg N/m²/yr, which we convert to mmol NO₃⁻/m³/yr via dimensional analysis and by dividing by the depth of the surface box. This source term of N to the model is independent of the modeled N terms.

**N₂ fixation**

N₂ fixation is the other source of new N to the model, and is assumed to only occur in the top box of the model. It is parameterized similarly to N₂ fixation in the model of DeVries et al. (2013), with partial inhibition by NO₃⁻ (Holl and Montoya, 2005) and dependence on iron (Fe) and phosphate (PO₄³⁻) availability (Monteiro et al., 2011).

7.   $J_{14}^{fix} = r_{14}^{fix} F_0 \ e^{-NO_{3,obs}/\lambda} \ e^{\frac{T_{obs}-T_{max}}{T_0}} \frac{Fe}{Fe+K_{Fe}} \frac{PO_4}{PO_4+K_P}$

$F_0$ is the maximum rate of N₂ fixation (1.5 mmol/m³/yr; Table 1) and is calculated from the estimated areal rate of N₂ fixation in the western tropical Atlantic (Capone et al., 2005) divided by the depth of the top model box. $NO_{3,obs}$ is the 2013 World Ocean Atlas (WOA) annually averaged surface NO₃⁻ interpolated to the model grid (Garcia et al., 2014). $\lambda$ is an inhibition constant for N₂ fixation in the presence of NO₃⁻.



The temperature (T) terms scale the rate of $N_2$ fixation based on the observed temperature ($T_{obs}$), maximum observed sea surface temperature ($T_{max}$), and the minimum preferred growth temperature for *Trichodesmium* ($T_0$; Capone et al., 2005). The temperature data were taken from 2013 WOA annually averaged temperature interpolated to the model grid (Locarnini et al., 2013).

Fe is the modeled deposition of soluble Fe interpolated to the model grid (mmol Fe/m$^2$/yr; Chien et al., 2016) divided by the depth of the top model grid box to give units of mmol Fe/m$^3$/yr. Fe and $PO_4^{3-}$ are assumed to limit $N_2$ fixation at low concentrations via Michaelis-Menten kinetics. $K_{Fe}$ and $K_P$ are their respective half-saturation constants. Additionally, there is a term that allows us to set the isotopic ratio of newly fixed N, $r_{14}^{fix}$, which is the fractional abundance of $^{14}N$ in newly fixed N

and is calculated as in Equation 6 from $\delta^{15}N_{fix}$ (-1‰; Table 1). All of the $N_2$ fixation parameters are fixed rather than optimized (Table 1). Due to the use of non-optimized parameters and an input $NO_3^-$ field rather than modeled $NO_3^-$, $N_2$ fixation serves as an independent check that our modeled N cycle produces reasonable N concentrations and overall N loss rates. However, $N_2$ fixation is not explicitly modeled here and is instead taken as a fixed, though spatially variable, input field.

In the model, $N_2$ fixation and $NO_3^-$ assimilation (Section 2.3.3) are assumed to be the two processes that create exportable OM. A fraction, σ, of this OM is portioned into DON rather than PON (Equations 3-4). In order to create spatial variability in this constant, we assumed (1-σ), the fraction of assimilated N partitioned to PON, is equal to the particle export (pe) ratio. This pe ratio is the ratio of particle export to primary production, and is equivalent to the fraction of OM that is exported from the euphotic zone as particulate matter rather than recycled or solubilized into DON. The pe ratio is calculated for each model grid

square from the mixed layer temperature ($T_{ml}$) and net primary production (NPP) as described by Dunne et al. (2005):

8.   $pe = \phi T_{ml} + 0.582 \log(NPP) + 0.419$

φ has a constant value of -0.0101 °C$^{-1}$. NPP estimates in units of mmol carbon/m$^2$/yr were taken from a satellite-derived productivity model (Westberry et al., 2008), annually averaged, and interpolated onto the model grid. $T_{ml}$ is calculated from the 2013 WOA annual average (Locarnini et al., 2013), which has been interpolated to the model grid. The temperature of the

top two model boxes were averaged to give $T_{ml}$. As temperature increases, the pe ratio decreases and less PON is exported, resulting in more DON recycling in the surface with several possible explanatory mechanisms discussed in greater detail by Dunne et al. (2005). As NPP increases, the pe ratio increases and more PON is exported; NPP explains 74% of the observed variance in particle export (Dunne et al., 2005).

### 2.3.3 Internal N cycling processes

**Assimilation of nitrate and nitrite**

Assimilation accounts for the uptake of DIN and its incorporation into OM. Since assimilation affects both the ON and DIN pools, we must account for it in both sets of model runs. Since the ON model is run first and the assimilation rates are dependent





on DIN concentrations, assumptions must be made about the DIN field in order to account for assimilation prior to the DIN model runs. We will first address assimilation in the ON equations.

The assimilation rates for DON and PON must be calculated using observed surface $[NO_3^-]$, rather than modeled $[NO_3^-]$. For this assumption to be valid, our modeled surface $[NO_3^-]$ must be close to the observed values, which we will test in Section 3.1. We also assume from the perspective of ON that only $NO_3^-$ is being assimilated, since $NO_2^-$ is present at relatively low concentrations in the surface ocean, and it may be characterized as recycled production. Assimilated N is partitioned between PON and DON using the pe ratio as previously described and shown in Equations 3 and 4.

$$9. \quad J_{14}^{assim} = {}^{14}k_{assim}[NO_3^-]_{obs}$$

$$10. \quad {}^{14}k_{assim} = \frac{NPP}{r_{C:N}[NO_3^-]_{obs}}$$

$[NO_3^-]_{obs}$ is the 2013 WOA annually averaged surface $[NO_3^-]$ interpolated to the model grid (Garcia et al., 2014). The rate constant for assimilation, ${}^{14}k_{assim}$, varies spatially and is determined using observations of surface $[NO_3^-]$ and satellite derived net primary production (NPP) estimates (Westberry et al., 2008). The rate constant is converted to N units using the ratio of carbon (C) to N in organic matter ($r_{C:N}$) which we assume to be 106:16 (Redfield et al., 1963). The value of the rate constant is only non-zero in the top two boxes of the model, where we assume primary production to be occurring. The same rate constant is used in both the ON and DIN assimilation equations.

The setup for assimilation in the DIN equations is similar, but can use modeled $[NO_3^-]$ and $[NO_2^-]$ rather than the WOA values. In order to appropriately reflect surface $NO_3^-$ and $NO_2^-$ concentrations, both $NO_3^-$ and $NO_2^-$ are assimilated. ${}^{14}k_{assim}$ is calculated as described above and is assumed to be the same for both $NO_3^-$ and $NO_2^-$. We justify using only $[NO_3^-]$ to parameterize ${}^{14}k_{assim}$ because $NO_3^-$ generally makes up the bulk of DIN available for assimilation at the surface, but this assumption will be discussed in more detail below.

$$11. \quad J_{14}^{assim,NO3} = {}^{14}k_{assim}[{}^{14}NO_3^-]$$

$$12. \quad J_{14}^{assim,NO2} = {}^{14}k_{assim}[{}^{14}NO_2^-]$$

**Solubilization**

Solubilization is the transformation of PON to DON, and is dependent only on [PON] and a solubilization rate constant (${}^{14}k_{sol}$), which is optimized (Table 2).

$$13. \quad J_{14}^{sol} = {}^{14}k_{sol}[PO^{14}N]$$

The solubilization of PON, together with the particle transport operator ($T_p$), produces a particle flux attenuation curve similar to a Martin curve with exponent b = -0.858 (Table 1) (Martin et al., 1987). While in the real world, the length scale for particle flux attenuation is somewhat longer in ODZs compared to oxygenated portions of the water column varies regionally (Berelson





et al., 2002; Buesseler et al., 2008; Buesseler and Boyd, 2009), our model uses a spatially-invariant $^{14}k_{sol}$. This is a refinement that could be introduced in future model versions.

**Remineralization**

Remineralization, or ammonification, is the release of DON into the DIN pool. This is determined using the concentration of DON and a remineralization rate constant ($^{14}k_{remin}$), which is optimized (Table 2).

$$14.\quad J_{14}^{remin} = {}^{14}k_{remin}[DO^{14}N]$$

The removal of this remineralized DON, since it does not accumulate as $NH_4^+$, is either through ammonia oxidation (AMO) or anammox (AMX), depending on $[O_2]$ as described below and in Section 2.3.4. We use the same remineralization rate

constant regardless of the utilized electron acceptor (e.g. $O_2$, $NO_3^-$). Since particle flux attenuation is observed to be somewhat weaker in oxygen deficient zones compared with oxygenated water (Van Mooy et al., 2002), this may slightly overestimate the rates of heterotrophic remineralization occurring in ODZs.

**Ammonia oxidation**

Ammonia oxidation (AMO) uses ammonia ($NH_3$) as a substrate. Since we do not include $NH_3$ or $NH_4^+$ in the model system, we treat remineralized DON as the substrate for AMO. In order to maintain consistency between the ON and DIN model runs, remineralized DON is routed either to AMO or AMX (lost from the system) based on the $O_2$ dependencies of AMO and AMX. Rather than using a strict $O_2$ cutoff for AMO, it is limited by $O_2$ using Michaelis-Menten kinetics. The half-saturation constant for $O_2$, $K_m^{AMO}$ (Table 1), sets the $O_2$ concentration at which AMO reaches half of its maximal value.

$$15.\quad J_{14}^{AMO} = (1 - \eta_{AMX})J_{14}^{rem} + \eta_{AMX}\frac{[O_2]}{[O_2] + K_m^{AMO}}J_{14}^{remin}$$

Recent studies have shown that AMO and $NO_2^-$ oxidation (NXR), both $O_2$-requiring processes, have very low $O_2$ half saturation constants and can occur down to nM levels of $[O_2]$ (Peng et al., 2015; Bristow et al., 2016b). In contrast, $O_2$-inhibited processes such as AMX are only allowed to occur at $O_2$ concentrations below a given threshold. The handling of $O_2$ thresholds for anaerobic processes is discussed in more detail below (Section 2.3.4). Briefly, the $O_2$ dependence of AMX is represented by

the parameter $\eta_{AMX}$, which has a value between 0 and 1 for a given grid box depending on the average number of months in a year its WOA 2013 falls below the $[O_2]$ threshold for anammox ($O_2^{AMX}$, Table 1). If, for example, $[O_2]$ in a given grid box is always above the threshold for AMX, $\eta_{AMX} = 0$ and all of the remineralized DON (represented by $J_{14}^{rem}$) will be oxidized via AMO. If $[O_2]$ is less than $O_2^{AMX}$, $\eta_{AMX}$ will be non-zero and a smaller fraction of the remineralized DON will be oxidized via AMO. The fraction ultimately oxidized by AMO is thus determined by the Michaelis-Menten parameterization of AMO, as

well as the $O_2$ threshold for anammox.





### Nitrite oxidation

The rates of $NO_2^-$ oxidation (NXR) are dependent on the availability of $NO_2^-$ as well as $O_2$. Similar to AMO, we parameterize $O_2$ dependence using Michaelis-Menten kinetics and a fixed half-saturation constant for $O_2$ ($K_m^{NXR}$, Table 1). $K_m^{NXR}$ was taken to be 0.8 μM $O_2$, based on kinetics experiments performed with natural populations of $NO_2^-$ oxidizing bacteria (Bristow et al.,

2016b). Finally, we employ an optimized rate constant ($^{14}k_{NXR}$, Table 2) to fit the available data.

16. $J_{14}^{NXR} = {}^{14}k_{NXR}[^{14}NO_2^-]\frac{[O_2]}{[O_2]+K_m^{NXR}}$

### 2.3.4 N sink processes

### Nitrate and nitrite reduction

$NO_3^-$ reduction (NAR) and $NO_2^-$ reduction (NIR) are two processes within the stepwise reductive pathway of canonical

denitrification. The end result of denitrification is the conversion of DIN as $N_2$ gas, rendering it bioavailable to only a restricted set of marine organisms. Although there are intermediate gaseous products between $NO_2^-$ and $N_2$, we treat $NO_2^-$ reduction as the rate-limiting step in the denitrification pathway, where DIN is removed from the system.

For both NAR and NIR, we introduce a dependency on two state variables. Since both NAR and NIR are heterotrophic

processes, they consume organic matter in addition to their main N substrates/electron receptors. In order to maintain levels of NAR and NIR that are bounded both by the available $NO_3^-$ or $NO_2^-$ and the available organic matter in a linear model, it was necessary to run ON and DIN equations separately, since it is not possible to include dependencies on two state variables (e.g. DON and $NO_3^-$) in the linear system. Both NAR and NIR are dependent on the remineralization rate ($J_{14}^{remin}$) that is calculated in the ON model run. In model boxes where NAR and NIR are occurring, some of the remineralization is carried out with

electron acceptors other than $O_2$. For simplicity, we assume that $J_{14}^{remin}$ does not depend on the choice of electron acceptor.

17. $J_{14}^{NAR} = \eta_{NAR}\,{}^{14}k_{NAR}[^{14}NO_3^-]\,J_{14}^{remin}$

18. $J_{14}^{NIR} = \eta_{NIR}\,{}^{14}k_{NIR}[^{14}NO_2^-]\,J_{14}^{remin}$

The rate coefficients for NAR ($^{14}k_{NAR}$) and NIR ($^{14}k_{NIR}$) are optimized rather than fixed (Table 2). Further, the dependence of $J_{14}^{NAR}$ and $J_{14}^{NIR}$ on $J_{14}^{remin}$ means that $k_{NAR}$ and $k_{NIR}$ are not first order rate constants and have different units than $k_{PON}$, $k_{DON}$,

and $k_{NXR}$ (Table 2).

The inhibition of NAR and NIR by $O_2$, like AMX, is parameterized by a parameter η, which inhibits these processes when $[O_2]$ is above their maximum threshold. Originally, we treated this term as a binary operator where it would be 0 if the empirically-corrected 2013 WOA annually averaged $[O_2]$ is above the threshold for the process and 1 if $[O_2]$ is below the

threshold. On further refinement, we wanted to account for the possibility of seasonal shifts in $[O_2]$ in ODZs. Thus, for each month, we assigned a value of 0 or 1 to each model grid box. These values were then averaged over the 12 months of the year to give a sliding value of η between 0 and 1 for each grid box. The $O_2$ thresholds used to calculate $\eta_{NAR}$ and $\eta_{NIR}$ were fixed



(7 μM and 5 μM, respectively; Table 1). Since we do not explicitly model $O_2$, $[O_2]$ was predetermined using the 2013 WOA monthly $O_2$ climatology (Garcia et al., 2014) interpolated to the model grid. We also applied an empirical correction that improves the fit of WOA $[O_2]$ data to observed suboxic measurements (Bianchi et al., 2012).

**Anammox**

Anammox (AMX) catalyzes the production of $N_2$ from $NH_4^+$ and $NO_2^-$. Since we do not use $NH_4^+$ as a variable in our N cycling equations, we substituted remineralized DON ($J_{14}^{remin}$) as a proxy for $NH_4^+$ availability. As described above in Section 2.3.3, remineralized DON is routed through either AMO or AMX depending on $[O_2]$ and the $O_2$ dependencies of AMO and AMX.

$$19. \quad J_{14}^{AMX} = \eta_{AMX}(1 - \frac{[O_2]}{[O_2]+K_m^{AMO}}) J_{14}^{remin}$$

The $O_2$ threshold used to calculate $\eta_{AMX}$ from monthly $O_2$ climatology is fixed (10 μM; Table 1). In order to maintain mass balance on remineralized DON, we do not include dependence on $[NO_2^-]$ in the rate equation 19, although $J_{14}^{AMX}$ removes $NO_2^-$ (equation 2). This parameterization inherently assumes that AMX is limited primarily by $[NH_4^+]$ supply and not $[NO_2]$, which may not always be correct (Bristow et al., 2016a). Anammox also produces 0.3 moles of $NO_3^-$ via associated NXR for every 1 mole of $N_2$ gas produced (Strous et al., 1999). For this reason, anammox appears in the state equation for $NO_3^-$ (Equation 1).

**Sedimentary denitrification**

Sedimentary (or benthic) denitrification ($J_{14}^{sed}$) is an important loss term for N in marine sediments, and in order to encapsulate it within the model grid we assume that it is occurring within the bottom depth box for any particular model water column. The parameterization for sedimentary denitrification is based on a transfer function as described by Bohlen et al. (2012). The

20 original transfer function was dependent on bottom water $[O_2]$, bottom water $[NO_3^-]$, and the rain rate of particulate organic carbon (RRPOC).

Here, RRPOC was calculated via a Martin curve (Martin et al., 1987) using the pe ratio, NPP, depth (z), euphotic zone depth ($z_{eu}$), and a Martin curve exponent (b):

$$20. \quad RRPOC = NPP * pe * (\frac{z}{z_{eu}})^b$$

NPP is derived from the productivity modeling of Westberry et al. (2008) as described in Section 2.3.2. The pe ratio is calculated as previously described in Section 2.3.2. The depth for any given model box is assumed to be the depth at the bottom of the box. The euphotic zone depth is the bottom depth of the 2nd box (73 m), since all production is assumed to be occurring in the top two boxes. As described above, the Martin curve exponent, b, is a fixed value in our model (b = -0.858; Table 1),

though this may result in underestimation of the particulate matter reaching the seafloor below ODZs (Van Mooy et al., 2002).

The transfer function for sedimentary denitrification was originally described using a non-linear dependence of the rate on ($[O_2] - [NO_3^-]$). In order for sedimentary denitrification to be properly implemented in our linear model, we broke the original




non-linear relationship into three roughly linear segments. We obtained three linear relationships between ($[O_2] - [NO_3^-]$) and sedimentary denitrification rate, each applicable across a given range of ($[O_2] - [NO_3^-]$) values (Figure S1). Due to the nature of our linear model, we needed to express our cutoff points between the segments in terms of $O_2$ rather than ($[O_2] - [NO_3^-]$). Therefore, a linear relationship between $O_2$ and ($[O_2] - [NO_3^-]$) was determined using the 2013 WOA annually averaged data

(Garcia et al., 2014; Figure S2). The cutoff points were determined to be 75 μM $O_2$ and 175 μM $O_2$. The linear relationships were then rearranged in order to estimate sedimentary denitrification rate as a function of RRPOC, $[O_2]$, and $[NO_3^-]$. These equations were then further broken down into a component that is dependent on $[NO_3^-]$ and a component that is dependent on $[O_2]$. In organic-rich shelf sediments an additional term is introduced that reduces the sedimentary denitrification rate by a globally averaged 27% if the depth of the bottom model box is less than 1000m due to the potential for efflux of $NH_4^+$ into the

water column (Bohlen et al., 2012). This decreases overall sedimentary denitrification by approximately 6 Tg N/yr. This transfer function also assumes that all of the $NH_4^+$ efflux is immediately oxidized to $NO_3^-$ and does not alter its isotopic composition in bottom water.

**2.4 N isotope implementation**

In our model, we are interested in using the isotopic composition of $NO_3^-$ and $NO_2^-$ to constrain the rates of N cycling and loss

from the global ocean. As DON and PON are ultimate substrates for $NO_2^-$ and $NO_3^-$ production, it is essential to track the $^{15}$N in the ON pools as well. The matrix setup is similar to that for the $^{14}$N species, but the rates were changed as follows:

21. $J_{15}^{process} = 1/\alpha_{process} \frac{[^{15}N_{substrate}]}{[^{14}N_{substrate}]} J_{14}^{process}$

$J_{14}^{process}$ is the rate of each relevant $^{14}$N process as described above, and $J_{15}^{process}$ is the rate of each $^{15}$N process. $\alpha_{process}$ is the fractionation factor for a given process, which is given by the ratio between the rate constants for $^{14}$N and $^{15}$N ($\alpha = {}^{14}k/{}^{15}k$). A

20 fractionation factor greater than 1 indicates a normal isotope effect and a fractionation factor less than 1 indicates an inverse isotope effect. Several of these fractionation factors are well known, but others are more poorly constrained, especially when values are calculated from *in situ* concentration and isotope ratio measurements (Hu et al., 2016; Casciotti et al., 2013; Ryabenko et al., 2012). For this reason, we ran several model cases with different fractionation factors for NAR, NIR, and NXR during the optimization process (Section 2.6, Table 3). The other fractionation factors were fixed (Table 1). In order to

25 produce the $^{15}$N concentrations of N species from our observations to constrain the model, we calculated $^{15}$N/$^{14}$N from measured $\delta^{15}$N, and assumed that $[^{14}N] \sim [^{14}N] + [^{15}N]$, the measured concentration of the modeled N species.

This simple $^{15}$N implementation was used with fixed fractionation factors for remineralization ($\alpha_{remin} = 1$), solubilization ($\alpha_{sol} = 1$), assimilation ($\alpha_{assim} = 1.004$), sedimentary denitrification ($\alpha_{sed} = 1$), and AMO ($\alpha_{AMO} = 1$) (Table 1). Isotope effects for

NAR ($\varepsilon_{NAR}$), NIR ($\varepsilon_{NIR}$), and NXR ($\varepsilon_{NXR}$) were varied in different combinations during model optimization (Table 3). Distinct isotopic parameterizations were also required for atmospheric deposition, $N_2$ fixation and anammox, as described below.





### Atmospheric deposition

For atmospheric deposition of N, we prescribe a constant $\delta^{15}$N value of -4‰ (Table 1), which can be related to the fractional abundance of $^{14}$N, previously described in Section 2.3.2 as $r_{14}^{dep}$, and the fractional abundance of $^{15}$N ($r_{15}^{dep}$) in deposited N. We multiply $r_{15}^{dep}$ by S$^{dep}$, the estimated rate of total N deposition, to obtain $J_{15}^{dep}$.

### Nitrogen fixation

Similar to atmospheric deposition, newly fixed N has a constant prescribed $\delta^{15}$N value (-1‰; Table 1). In Section 2.3.2 we described $r_{14}^{fix}$, the fractional abundance of $^{14}$N in newly fixed N. Here we multiply the fractional abundance of $^{15}$N, $r_{15}^{fix}$, by the other terms in the N$_2$ fixation equation (Equation 6) to obtain the rate of $^{15}$N fixation.

### Anammox

Anammox is the most complicated to parameterize isotopically because it has three different isotope effects associated with it. There is an isotope effect on both substrates converted to N$_2$ (NO$_2^-$ and NH$_4^+$), as well as for the associated NO$_2^-$ oxidation to NO$_3^-$. We assume that the fractionation factor for ammonium oxidation via AMX ($\alpha_{AMX,NH4}$) is 1, setting it to match the

15 fractionation factor for AMO ($\alpha_{AMO}$; Table 1), both with no expressed fractionation since NH$_4^+$ does not accumulate in the model. Since all remineralized DON must be routed either through AMO or AMX, this simplifies the mass balance and ensures that all remineralized $^{14}$N and $^{15}$N is accounted for. $^{15}$NO$_2^-$ is removed with the isotope effects of NO$_2^-$ reduction ($\alpha_{AMX,NIR}$) and NO$_2^-$ oxidation ($\alpha_{AMX,NXR}$), in the expected 1:0.3 proportion (Brunner et al., 2013).

### 2.5 Model inversion

Once our N cycle equations were set up as described above, we input them into MATLAB in block matrix form. The equations were of the general form $Ax = b$. All 200,160 model boxes are accounted for in the matrices. Matrix $A$ (400,320 x 400,320) contained the rate constants and other parameters that are multiplied by the vector of state variables, $x$ (400,320 x 1). Vector $x$ contained the state variables (i.e. [NO$_3^-$] and [NO$_2^-$] or [DON] and [PON]) to be solved for by the linear solver. Vector $b$ (400,320 x 1) contained the rates that were independent of the state variables, such as N$_2$ fixation and N deposition. Let us

consider, as an example, the DIN model setup. The top left corner of matrix $A$ would contain rate constants for processes that produce and consume NO$_3^-$ that are also dependent on [NO$_3^-$]. The top right corner of matrix $A$ would contain rate constants for processes that produce and consume NO$_3^-$ but are dependent on [NO$_2^-$]. The bottom left corner of matrix $A$ would contain rate constants for processes that produce and consume NO$_2^-$ but are dependent on [NO$_3^-$]. The bottom right corner of matrix $A$ would contain rate constants that produce and consume NO$_2^-$ and are also dependent on [NO$_2^-$]. The top half of vector $x$ would

be [NO$_3^-$] for each model box, and the bottom half of vector $x$ would be [NO$_2^-$] for each model box. The top half of vector $b$ would be independent processes that produce or consume [NO$_3^-$], and the bottom half of vector $b$ would be independent processes that produce or consume [NO$_2^-$].





In MATLAB, we used METIS ordering, which is part of SuiteSparse (http://faculty.cse.tamu.edu/davis/suitesparse.html) to order our large, sparse matrix $A$. We then used the built-in function `umfpack` with METIS to factorize matrix $A$. The built-in matrix solver `mldivide` was then used with the factorized components of matrix $A$ and matrix $b$ to solve for $x$.

**2.6 Parameter optimization**

There are many parameters in the model that control the rates of the different N cycle processes (Tables 1-3). Some of these parameters are well constrained by literature values. Others, such as the rate constants, were objects of our investigation and were optimized against available observations. For our optimization, we compared model output using different parameter values to a database of $NO_3^-$ and $NO_2^-$ concentrations and isotopes. The database was originally compiled by Rafter et al. (in

prep.) and has been expanded to include some additional unpublished data (Table S1). All of the database observations were binned and interpolated to the model grid. If multiple observations occurred within the same model grid box, the values were averaged and a standard deviation was calculated. The database was divided randomly into a training set, used for optimization, and a test set, used to assess model performance. The same number of grid points with observations was used in the training and test sets.

The optimization procedure used the MATLAB function `fminunc` to obtain values for the non-fixed parameters that minimized a cost function. In each iteration of the optimization, the model system was solved by running the $^{14}$N-ON model, $^{15}$N-ON model, $^{14}$N-DIN model, and $^{15}$N-DIN model. The modeled output $[NO_3^-]$, $[NO_2^-]$, $\delta^{15}N_{NO3}$, and $\delta^{15}N_{NO2}$ were compared to values from the database training set. Though DON and PON observations were not used to optimize the model, the open

ocean and deep water $NO_3^-$ values were useful in constraining the parameters that control PON solubilization and DON remineralization. The entire model was run using a set of initial parameter values (Table 2) and the optimization scheme continued to alter those starting parameters until a minimum in the cost function was attained. We optimized the logarithm of the parameter values rather than the original parameters themselves so the unconstrained optimization returned positive values. The transformed starting parameters and subsequent modified parameter sets were then fed back into the model equations as

$e^x$, where $x$ denotes the log-transformed parameter. The cost function in the optimization procedure is as follows:

22.  $$\text{Cost} = \frac{w_{NO3}}{n_{NO3}sd_{NO3}}\sum([NO_3^-]_{model} - [NO_3^-]_{training})^2 + \frac{w_{NO2}}{n_{NO2}sd_{NO2}}\sum([NO_2^-]_{model} - [NO_2^-]_{training})^2 +$$
$$\frac{w_{\delta NO3}}{n_{\delta NO3}sd_{\delta NO3}}\sum(\delta^{15}N_{NO3,model} - \delta^{15}N_{NO3,training})^2 + \frac{w_{\delta NO2}}{n_{\delta NO2}sd_{\delta NO2}}\sum(\delta^{15}N_{NO2,model} - \delta^{15}N_{NO2,training})^2$$

The $w$ terms are weighting terms intended to scale the contributions of the four observed parameters in order to equalize their contributions to the cost function. The $n$ terms and standard deviation ($sd$) terms were used to normalize the contributions of

each measurement type to the cost function. Each $n$ term is equal to the number of each type of measurement in the training data set (e.g. the number of $[NO_3^-]$ data points = $n_{NO3}$). The $sd$ term is equal to the standard deviation of all the measurements of a given type (e.g. the standard deviation of all the $[NO_3^-]$ data points within the training set).



In order to account for error in our model parameter estimates, we also iterated over several possible values for three of the most important isotope effects for processes in ODZs: NAR, NIR, and NXR. We chose to iterate over these parameters rather than optimize them since there is a large range of estimates from the literature as to what these parameters might be (Table 3).

We assigned different possible values for each of these parameters (Table 3), resulting in 12 possible combinations. The optimization protocol was performed for each of those combinations and unique optimized parameter sets were obtained. The parameter results were then averaged (final values, Table 2) and their spread is categorized as the error (error, Table 2).

### 3 Results

### 3.1 Global model-data comparison

The simulations of $NO_2^-$ distribution and its isotopic composition are the most unique features of this model in comparison to many other existing global models of the marine N cycle. As such, $NO_2^-$ accumulation in ODZs is a feature that should be well-represented by the model in order to use it to test hypotheses about processes that control N cycling and loss in ODZs. Overall, we see $NO_2^-$ accumulating at 200m in the major ODZs of the Eastern Tropical North Pacific (ETNP), Eastern Tropical South Pacific (ETSP), and the Arabian Sea (AS) (Figure 2), which is consistent with observations and expected based on the

low $O_2$ conditions found there. However, accumulation of $NO_2^-$ in the model ETSP was lower than expected. The model also accumulated $NO_2^-$ in the Bay of Bengal, which is a low-$O_2$ region off the east coast of India that does not generally accumulate $NO_2^-$ or support water column denitrification, but is thought to be near the "tipping point" for allowing N loss to occur (Bristow et al., 2016a). The Bay of Bengal will be discussed further in Section 4.2.

The model optimization described above yielded a set of isotope effects that best fit the global dataset of $[NO_3^-]$, $[NO_2^-]$, $\delta^{15}N_{NO3}$ and $\delta^{15}N_{NO2}$. The best fit was achieved for isotope effects of 13‰ for $NO_3^-$ reduction ($\varepsilon_{NAR}$), 0‰ for $NO_2^-$ reduction ($\varepsilon_{NIR}$), and -13‰ for $NO_2^-$ oxidation ($\varepsilon_{NXR}$). Figure 3 shows the test set comparison for the global best-fit set of isotope effects overlaid with a 1:1 line, which the data would follow if there was perfect agreement between model results and observations. There is general agreement between model and observations, with most of the data clustering near the 1:1 lines. Agreement

between the observations and the training data are similar (Figure S3), indicating that we did not overfit the training data.

In the test set, there were some low $[O_2]$ points where our model $[NO_3^-]$ exceeded observations (Figure 3a, filled black circles); these are largely within the Eastern Tropical South Pacific (ETSP), where insufficient $NO_3^-$ reduction occurred in the model. In contrast, the Arabian Sea tended to show slightly lower modeled $[NO_3^-]$ than expected. In addition, the $[NO_2^-]$ accumulation

(Figure 3b) and $\delta^{15}N_{NO3}$ signals (Figure 3c) in the ETSP were generally too low compared with observations. These signals are likely tied to insufficient $NO_3^-$ reduction in the ETSP, although excessive consumption of $NO_2^-$ may also play a role.



Another consideration is that there may be a mismatch in resolution between the model and the space and time scales needed to resolve the high $NO_2^-$ accumulations observed sporadically (Anderson et al., 1982; Codispoti et al., 1985; 1986). Overall, the fits of $\delta^{15}N_{NO3}$ were fairly good (RMSE = 2.4‰), though there were some points above $\delta^{15}N_{NO3}$ = 10‰ where the modeled $\delta^{15}N_{NO3}$ exceeded the observed $\delta^{15}N_{NO3}$ and others where modeled $\delta^{15}N_{NO3}$ was lower than observations (Figure 3c). Many of

the points with overestimated $\delta^{15}N_{NO3^-}$ were located within the Arabian Sea ODZ, where there may be too much NAR occurring, leading to excess enrichment in $^{15}N\text{-}NO_3^-$. The fits of $\delta^{15}N_{NO2}$ were also fairly good (RMSE = 8.6‰), though the modeled $\delta^{15}N_{NO2}$ was generally not low enough (Figure 3d), indicating an underestimated sink of 'heavy' $NO_2^-$.

**3.2 Oxygen deficient zone model-data comparison using station profiles**

To further investigate the distribution of model N species within the three main ODZs, we selected representative grid boxes

within each ODZ that contained observations to directly compare with model results in station profiles. Overall, the modeled $NO_3^-$ and $NO_2^-$ concentration and isotope profiles in the AS and ETNP were consistent with the observations, with $[NO_3^-]$ slightly under estimated in the Arabian Sea ODZ and overestimated in the ETSP (Figure 4). As $[O_2]$ goes to zero, the $O_2$-intolerant processes NAR, NIR, and AMX are released from inhibition. These processes result in a decrease in $[NO_3^-]$ (via NAR) which corresponds to an increase in $\delta^{15}N_{NO3}$, since NAR has a normal isotope effect. $NO_2^-$ also starts to accumulate in

the SNM as a result of NAR. $\delta^{15}N_{NO2}$ is lower than $\delta^{15}N_{NO3}$ since light $NO_2^-$ is preferentially created via NAR, and this discrepancy is further reinforced by the inverse isotope effect of NXR (Casciotti, 2009). These patterns are readily observed in the AS and ETNP, but were less apparent in the ETSP, where $[NO_3^-]$ depletion and $[NO_2^-]$ accumulation in the model were lower than observed.

In order the gauge the model results for N loss, we also calculated N*, a measure of the availability of DIN relative to $PO_4^{3-}$ compared to Redfield ratio stoichiometry (N* = $[NO_3^-]$ + $[NO_2^-]$ – 16 * $[PO_4^{3-}]$; Deutsch et al., 2001). Negative N* values are associated with N loss due to AMX or NIR, while positive N* values are associated with input of new N through $N_2$ fixation (Gruber and Sarmiento, 1997). Although we did not model $PO_4^{3-}$, we used the modeled $[NO_3^-]$ and $[NO_2^-]$ together with WOA $PO_4^{3-}$ data interpolated to the model grid to calculate N* resulting from the model. Both the AS and ETNP showed a decrease

in model N* in the ODZ, as expected for water column N loss. Below the ODZ, N* increased again and returned to expected deep water values. Modeled N* in the ETSP, however, did not follow the observed trend, consistent with an underestimate of N loss in the model ETSP.

Though the global best fit isotope effects for NAR, NIR, and NXR produced good agreement to the data in general, the isotope

effects that best fit individual ODZ regions differed when the cost function was restricted to observations from a given ODZ. For the ETSP, the best fit isotope effects were the same as the previously stated global best fit. For the AS, the best fit isotope effects were $\varepsilon_{NAR}$ = 13‰, $\varepsilon_{NIR}$ = 0‰, and $\varepsilon_{NXR}$ = -32‰. For the ETNP, the best fit isotope effects were $\varepsilon_{NAR}$ = 13‰, $\varepsilon_{NIR}$ = 15‰, and $\varepsilon_{NXR}$ = -32‰, though the performance is only marginally better than with $\varepsilon_{NIR}$ = 0‰. The lower (more inverse) value



for $\varepsilon_{NXR}$ resulted in higher $\delta^{15}N_{NO3}$ and lower $\delta^{15}N_{NO2}$, which better fit the ODZ $\delta^{15}N_{NO2}$ data compared to the global best fit $\varepsilon_{NXR}$ = -13‰. These results are consistent with earlier isotope modeling studies in the ETSP (Casciotti et al., 2013; Peters et al., 2016; Peters et al., 2018b) and in the Arabian Sea (Martin and Casciotti, 2017). Although, in the Arabian Sea, modeled $\delta^{15}N_{NO3}$ were too high in the ODZ, likely in part due to overpredicted rates of NAR, which also resulted in lower modeled

[$NO_3^-$] (Figure 4).

### 3.3 Model-data comparison in GEOTRACES sections

We also investigated the agreement between global best fit modeled [$NO_3^-$] and $\delta^{15}N_{NO3}$ with data from two GEOTRACES cruise sections: GP16 in the South Pacific, and GA03 in the North Atlantic. For GP16, we see that [$NO_3^-$] matches fairly well in the surface waters, but diverged below 500m as well as at the eastern edge of the ETSP ODZ (Figure 5). Although the

patterns are generally correct, insufficient $NO_3^-$ is being accumulated in the deep waters of the model Pacific, which could be due to an underestimate of preformed $NO_3^-$ (over estimate of assimilation in the Southern Ocean), or inadequate supply of organic matter to be remineralized at depth. In the Southern Ocean, model surface [$NO_3^-$] are 5-10 μM lower than observations (Figure S4), which could be enough to explain the lower than expected [$NO_3^-$] in the deep Pacific, which is largely sourced from the Southern Ocean (Rafter et al., 2013; Sigman et al., 2009; Peters et al., 2018a,b).

In the GP16 section, we also see that the insufficient depletion of $NO_3^-$ in the ETSP ODZ (Figure 5b) extends beyond the single grid box highlighted earlier (Figure 4). Along with this, there is also minimal increase of $\delta^{15}N_{NO3}$ in the ETSP ODZ and the upper thermocline in the eastern part of the section, consistent with an underestimate of $NO_3^-$ reduction. Surface $\delta^{15}N_{NO3}$ values were also not as high in the model as in the observations (Figure 5), which could result from insufficient $NO_3^-$ assimilation or

too low supplied $\delta^{15}N_{NO3}$ (Peters et al., 2018a). However, we do see a similar depth range for high surface $\delta^{15}N_{NO3}$ and a local $\delta^{15}N_{NO3}$ minimum between the surface and ODZ propagating westward in both the model and observations, indicating that the physical processes affecting $\delta^{15}N_{NO3}$ are represented by the model. Additionally, the model shows slightly elevated $\delta^{15}N_{NO3}$ in the thermocline depths (200-500m) west of the ODZ, which is consistent with the observations (Figure 5c), though not of the correct magnitude. This is partly related to the muted ODZ signal as mentioned above and its lessened impact on thermocline

$\delta^{15}N_{NO3}$ across the basin. Peters et al. (2018a) and Rafter et al. (2013) also postulated that these elevated $\delta^{15}N_{NO3}$ values were in part driven by remineralization of organic matter with high $\delta^{15}N$. The $\delta^{15}N$ of sinking PON in the model (6-10‰) was similar to those observed in the south Pacific (Raimbault et al., 2008), as well as those predicted from other N isotope studies (Rafter et al., 2013; Peters et al., 2018). The model also shows slightly elevated $\delta^{15}N_{NO3}$ in the intermediate depths (500-1500m), which is consistent with observations, again reflecting remineralization of PON with $\delta^{15}N$ greater than mean ocean

$\delta^{15}N_{NO3}$. Overall, the patterns of $\delta^{15}N_{NO3}$ are correct but the magnitudes of isotopic variation are muted, largely due to the lack of N loss in the ODZ and modeled surface $\delta^{15}N_{NO3}$ values that are lower than observations.





In the north Atlantic along GEOTRACES section GA03, we see good agreement between the observed and modeled [NO$_3^-$] (Figure 6). There is generally low surface [NO$_3^-$] with a distinct area of high [NO$_3^-$] propagating from near the African coast. Deep water (> 2000 m) [NO$_3^-$] is lower than we see in the Pacific section, and the model matches well with the Atlantic

observations. Again, there is not quite enough NO$_3^-$ present in Southern Ocean-sourced intermediate waters (500-1500 m; Marconi et al., 2015). Modeled $\delta^{15}N_{NO3}$ values at first glance appear higher than observed values at the surface (Figure 6). However, many of the surface [NO$_3^-$] were below the operating limit for $\delta^{15}N_{NO3}$ analysis and were not determined. Focusing on areas where both measurements and model results are present yields excellent agreement. For example, we do see low $\delta^{15}N_{NO3}$ values in upper thermocline waters in both the model and observations, likely corresponding to low $\delta^{15}N$ contributions

from N$_2$ fixation that is remineralized at depth and accumulated in North Atlantic Central Water (Marconi et al., 2015; Knapp et al., 2008). The model input includes high rates of N$_2$ fixation in the North Atlantic that are consistent with this observation (Martin et al., in prep). However, rates of N deposition in the North Atlantic are also fairly high and can contribute to the low $\delta^{15}N$ signal (Knapp et al., 2008). In our model, atmospheric N deposition contributed between 0-50% of N input along the cruise track.

## 4 Discussion

### 4.1 Assumption checks

As previously mentioned (Section 2.3), ON and DIN were modeled separately in order to introduce dependence on both ON and substrate availability for the heterotrophic processes NAR and NIR. This separate modeling requires several assumptions to be made regarding the processes that impact both ON and DIN, namely assimilation and remineralization.

The first assumption is that the rates of N assimilation are equal between the ON and DIN model runs. The ON model run used WOA surface [NO$_3^-$] to estimate the contribution of DIN assimilation to the production of ON, whereas the DIN model run used modeled [NO$_3^-$] and [NO$_2^-$] to estimate DIN removal via assimilation. Though these two methods utilized the same rate constants for assimilation, differences in [DIN] could cause some discrepancies between the overall rates. Analysis of the

results revealed that slightly more overall DIN assimilation occurred in the DIN model run than ON produced in the organic N model (Figure S5). This could be due in part to assimilation of NO$_2^-$ in the top two boxes, since NO$_2^-$ assimilation is unaccounted for in the ON model. This is largely an issue in the oligotrophic gyres, where surface [NO$_3^-$] is very low and NO$_2^-$ accumulates to low but non-zero concentrations (Figure 2). Assimilation of NO$_2^-$ accounts for a significant fraction of DIN assimilation in these regions, but the overall assimilation rates there are low and the resulting influence on the whole system

is also low. In other regions, modeled surface [NO$_3^-$] may be higher than the WOA surface [NO$_3^-$] data that are supplied to the ON model, which would result in higher assimilation rates in the DIN model run. Indeed, points at which the DIN assimilation rates are higher than the ON production rates do tend to have modeled [NO$_3^-$] that was higher than observed [NO$_3^-$] (Figure



S5). Likewise, points with relatively lower DIN assimilation had modeled [$NO_3^-$] less than observed [$NO_3^-$]. However, the majority of DIN assimilation estimates were within 10 μM/yr of the ON production estimates, with an overall DIN assimilation rate of 9235 μM/yr. We also found that the WOA surface $NO_3^-$ values are fairly well represented by our modeled surface $NO_3^-$ (Figure S4). We conclude that though the assimilation rates are not identical in the ON and DIN model runs, the discrepancy

in modeled DIN assimilation is less than 0.1%, and there is unlikely to be significant creation or loss of N as a result of the split model.

### 4.2 Model dependency on input $O_2$

The modeled concentration and isotope profiles for the ETSP, unlike in the AS and ETNP, reflected a significant underestimation of water column denitrification in the best-fit model. In ETSP [$NO_3^-$] measurements, there is a clear deficit in

[$NO_3^-$], coincident with the SNM and N* minimum (Figure 4). In our modeled profiles, this $NO_3^-$ deficit is missing, and although a SNM is present, its magnitude is lower than observed (Figure 4). The model also does not capture the negative N* excursion (Figure 4), which we think reflects a model underestimation of $NO_3^-$ and $NO_2^-$ reduction in the ETSP. The cause of this missing denitrification is likely to be poor representation of the ETSP $O_2$ conditions in the model grid space. Since our model grid is fairly coarse (2˚x2˚), only a few boxes within the ETSP had averaged [$O_2$] below the thresholds that would allow

processes such as NAR and NIR to occur. The anoxic region of the ETSP is adjacent to the coast and not as spatially extensive as in the AS and ETNP (Figure S6); therefore, this region in particular was less compatible with the model grid. In order to test whether the parameterization of $O_2$ dependence was the cause of the low N loss, we ran the model using the globally optimized parameters (Table 3) but with higher $O_2$ thresholds (15 μM) for NAR, NIR, and AMX (Table 1). This extended the region over which ODZ processes could occur and resulted in an increase in water column N loss from 6 to 32 Tg N/yr in the

ETSP, which is more consistent with previous estimates (DeVries et al., 2012; Deutsch et al., 2001). This change also stimulated the development of a $NO_3^-$ deficit, larger secondary $NO_2^-$ maximum, and N* minimum within the ODZ (Figure 7).

As previously mentioned in Section 3.1, modeled [$NO_2^-$] in the Bay of Bengal is higher than observations. The accumulation of $NO_2^-$ here in the model is likely due to $O_2$ concentrations falling below the set threshold for NAR but above the threshold

for NIR. Although AMX and NXR occur there, the modeled rates are rather slow, which leads to more $NO_2^-$ being produced than consumed. This is in contrast to observations that $NO_2^-$ production via NAR is tightly matched with $NO_2^-$ consumption via NXR, which limits $NO_2^-$ accumulation and N loss in the Bay of Bengal (Bristow et al., 2016a). The fact that the model over predicts NAR in the Arabian Sea may also be connected with over-prediction of NAR in the Bay of Bengal, leading to accumulation of $NO_2^-$ despite realistic $NO_2^-$ consumption rates. Further work on oxygen sensitivities of N cycle processes will

be addressed in a companion study (Martin et al., in prep).



## 5 Conclusions

A global inverse ocean model modified to include $[NO_2^-]$, $\delta^{15}N_{NO3}$, and $\delta^{15}N_{NO2}$ as state variables, as well as adding the processes required to describe the cycling of $NO_2^-$ in the global ocean resulted in a globally representative distribution of $[NO_3^-]$, $[NO_2^-]$, $\delta^{15}N_{NO3}$, and $\delta^{15}N_{NO2}$. In particular, the patterns of variation in both oxic and anoxic waters are generally consistent

with observations, though some magnitudes of variation were somewhat muted by the model. This could be due to an underestimation of a process rate, due to parameterization or model resolution, or an underestimation of the isotope effect involved.

Importantly, we were able to generate a roughly balanced steady-state ocean N budget without the need for an artificial

restoring force. The $[NO_3^-]$ and $[NO_2^-]$ distributions that were required to achieve this roughly balanced budget are well within the range of observed values. Some interesting take-home messages from this work are 1) a relatively low isotope effect for $NO_3^-$ reduction ($\varepsilon_{NAR} = 13$‰) gives a good fit to $\delta^{15}N_{NO3}$ data, similar to that concluded in some recent studies (Marconi et al., 2017; Casciotti et al., 2013), 2) low $O_2$ half-saturation constants for $NO_2^-$ oxidation allowing NXR to occur in parallel with NAR, NIR, and AMX were needed to achieve the correct distributions of $NO_3^-$, $NO_2^-$, and their isotopes, in the oceans water

column ODZs.

Though we have been able to adequately represent and assess N cycling in ODZs, there are many areas in which this model could be improved in order to expand its usefulness. Improving resolution of the model, particularly in coastal regions where there are steep gradients in nutrient and $O_2$ concentrations, would improve the accuracy of the model in regions such as the

ETSP. Further, in regions that have high seasonal or interannual variability, an annually averaged steady-state model may not represent some important temporal dynamics. While we attempted to account for seasonal variation in the strength of the ODZs, we did not simulate seasonal variations in NPP and the strength of the biological pump.

In addition to the dependency on external static nutrient and parameter fields, this N cycle model is highly dependent on isotope

effects for N cycle processes. As discussed in Section 2.6, the rate parameters were optimized using a variety of different combinations of isotope effects for NAR, NIR, and NXR to explore model uncertainty. As presented in Section 3.1, the larger magnitude isotope effect for NXR best fit the ETNP and Arabian Sea ODZs, where most of the ODZ volume resided. The larger magnitude isotope effect also resulted in optimized rate constants for NAR, NIR, and NXR that were lower than the global best fit rate constants. The lower rate constants and larger isotope effects resulted in better fits to observations of $\delta^{15}N$

and DIN concentrations. This reinforces the importance of obtaining realistic isotope effect estimates for each process that are relevant on an environmental scale.

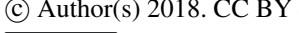


Previous work has shown that the laboratory-derived isotope effects for some N cycle processes are not the same as their expressed isotope effects in environmental samples or under conditions relevant to environmental samples (Casciotti et al., 2013; Bourbonnais et al., 2015; Buchwald et al., 2015; Kritee et al., 2012; Marconi et al., 2017a). Further probing the isotope effects using an inverse model such as this could provide insight into the expressed isotope effects that should be used in other

5 modeling efforts involving field data. Additionally, this reinforces the need for critical consideration of isotope effects used in N cycle models that use isotope balance to predict N cycling rates. Though isotopes provide us a useful tool to assess the relative contributions of different processes, these estimates are highly subject to the isotope effects employed. Also, as illustrated by the regional optimizations, the isotope effect for a given process may be vary, or be expressed differently, in different regions.

This model provides an excellent framework for testing hypotheses about controls on the marine N inventory and cycling of N on a global scale. The distribution of N cycle rates resulting from this model will be explored in a companion manuscript (Martin et al., in prep). Incorporation of modified environmental input data, such as temperature, productivity, and $[O_2]$, could also help us predict how the N cycle might be affected by past and future environmental changes.

**Code availability**

Model code and model output from the three optimal ODZ isotope effect combinations, including the global best fit, are available in the Stanford Digital Repository (https://purl.stanford.edu/hr045dx8661).

**Author contribution**

KLC, FP, and TSM designed the study. TSM and FP constructed the model. TSM and KLC analysed and interpreted the
20 results. TSM, KLC, and FP wrote the manuscript.

**Competing interests**

The authors declare that they have no conflict of interest.

**Acknowledgements**

Thanks to Patrick Rafter for sharing a pre-publication version of his $NO_3^-$ isotope database. Thanks to Tim Davis for guidance
on sparse matrix solvers. Thanks to Tim DeVries for helpful discussions about earlier versions of the inverse model. Thanks to Kevin Arrigo and Leif Thomas for comments on an earlier draft of this manuscript. This work was partly supported by NSF Chemical Oceanography grant 1657868 to KLC.




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



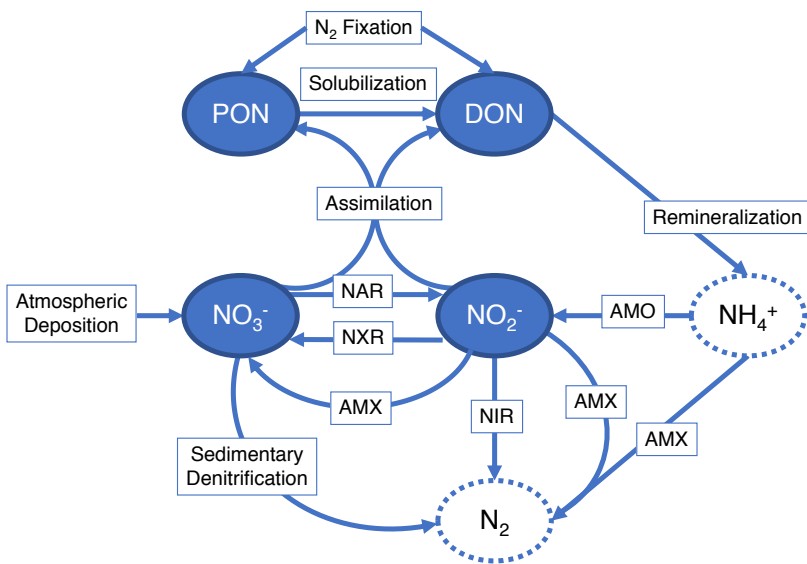

**Figure 1: Diagram showing the nitrogen (N) cycle processes represented in the model. Two organic N pools are modeled: particulate organic N (PON) and dissolved organic N (DON). Two inorganic N pools are modeled: nitrate ($NO_3^-$) and nitrite ($NO_2^-$). N source processes are nitrogen ($N_2$) fixation and atmospheric deposition. N sink processes are sedimentary denitrification, $NO_2^-$ reduction (NIR), and anammox (AMX). Internal cycling processes that transform N from one species to another are solubilization, remineralization, assimilation, $NO_3^-$ reduction (NAR), ammonia oxidation (AMO), and $NO_2^-$ oxidation (NXR). Neither ammonia ($NH_3$) nor ammonium ($NH_4^+$) are tracked in this model, since they are assumed to not accumulate. $N_2$ is also not explicitly accounted for in the model.**





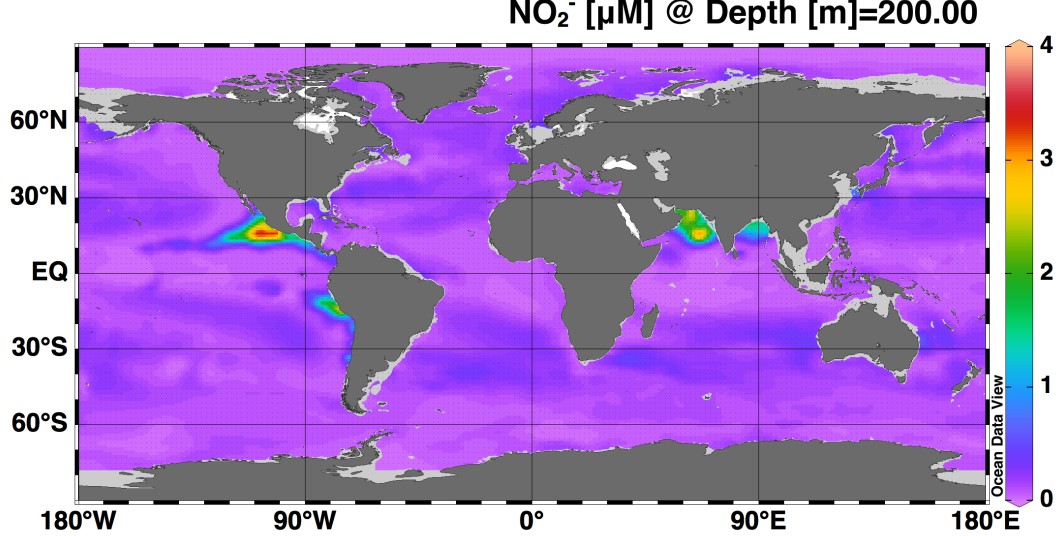

Figure 2: Map showing the model-estimated accumulation of nitrite ($NO_2^-$) at 200 m depth.





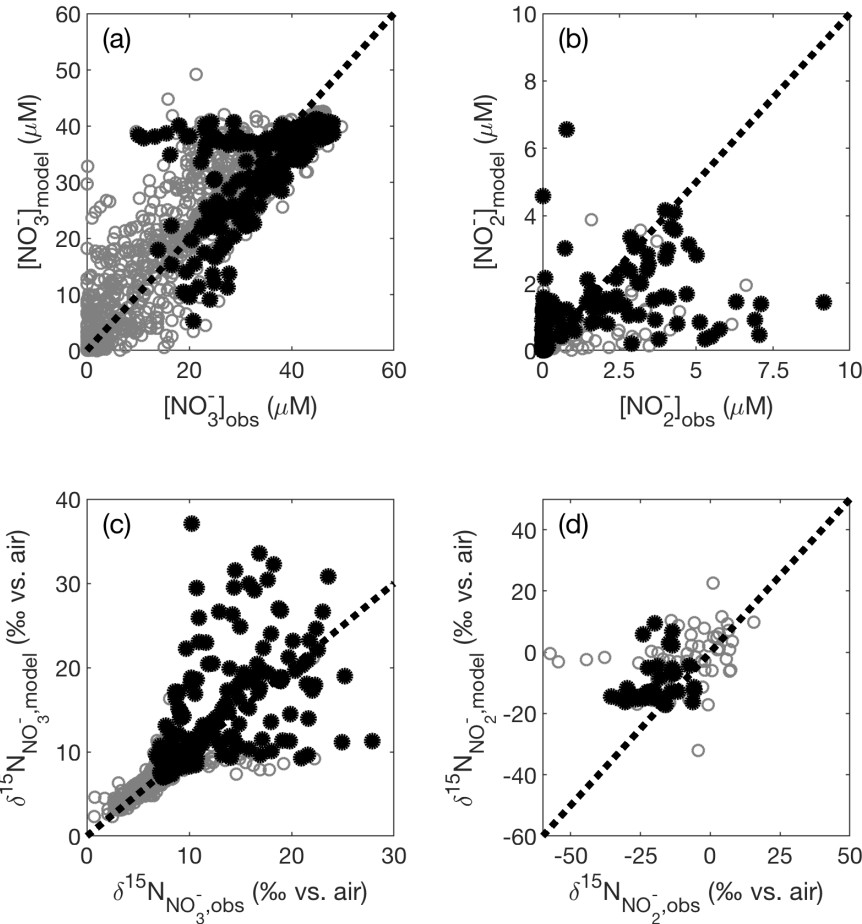

**Figure 3: Modeled (a) [NO₃⁻], (b) [NO₂⁻], (c) δ¹⁵N_NO3-, and (d) δ¹⁵N_NO2- are compared against the corresponding values from the database test set. Shown on each panel is a 1:1 line starting at the origin. Data in black have corresponding [O₂] < 10 μM, and data in gray have [O₂] ≥ 10 μM.**

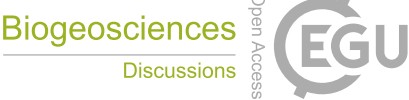



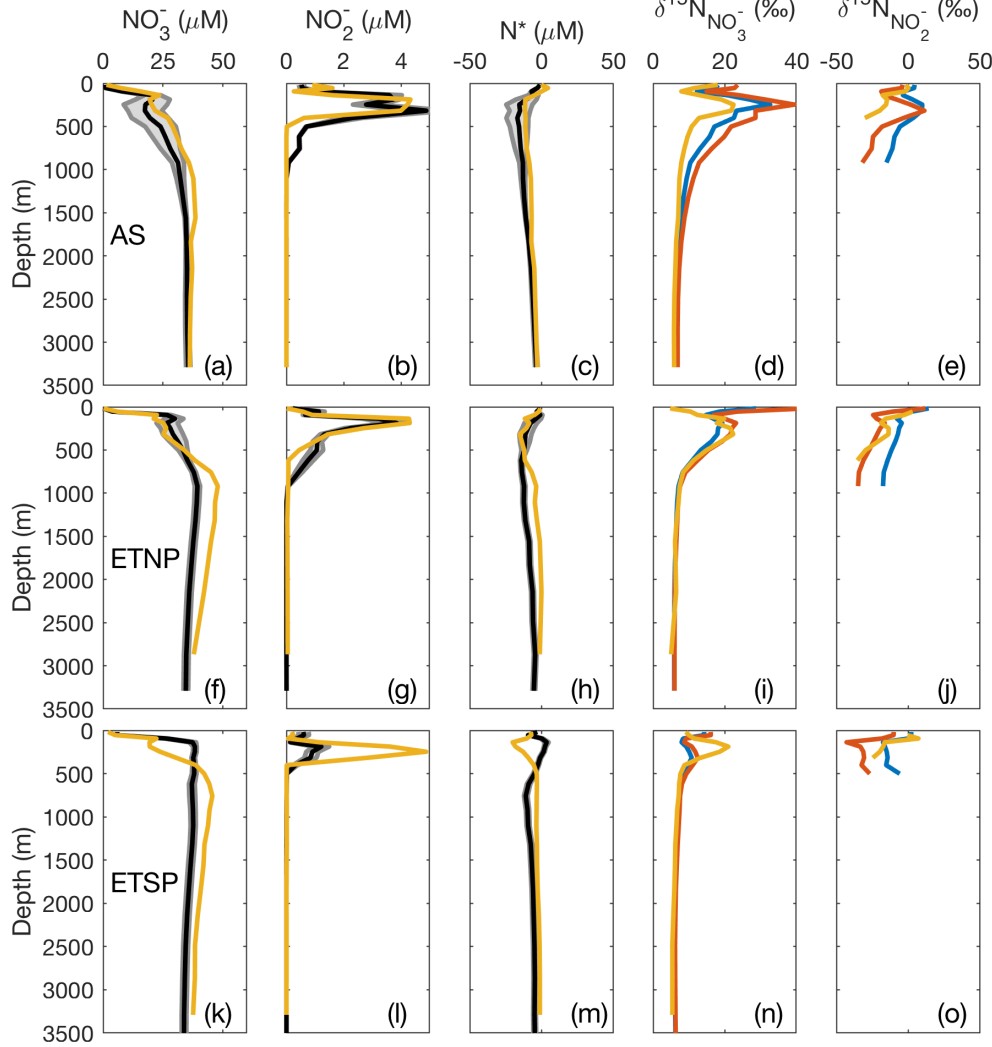

**Figure 4: Depth profiles comparing model results with binned and averaged database observations from a model water column. Results are shown for the three main oxygen deficient zones (ODZs): the Arabian Sea (a-e), the Eastern Tropical North Pacific (ETNP; f-j), and the Eastern Tropical South Pacific (ETSP; k-o). Average modeled nitrate concentration ([NO₃]), nitrite**

5   **concentration ([NO₂⁻]), and N\* are shown in black. Gray error lines around the black line show the 2σ spread from the average from the 12 different optimized model results using the different combinations of isotope effects for nitrate reduction (εNAR), nitrite reduction (εNIR), and nitrite oxidation (εNXR). Observed data are shown in yellow in all panels. Modeled δ¹⁵NNO3- and δ¹⁵NNO2- are shown for three different combinations of isotope effect. The blue lines represent εNAR = 13, εNXR = -13, and εNIR = 0, which are the best fit isotope effects globally and in the ETSP. The red lines represent εNAR = 13, εNXR = -32, and εNIR = 0, which are the best fit**

10   **isotope effects in the Arabian Sea.**





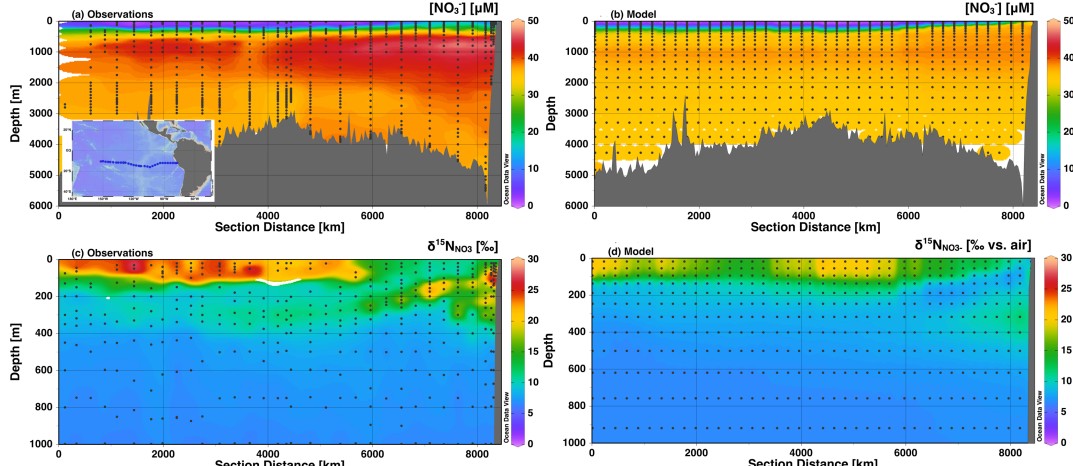

**Figure 5:** Section profiles of NO₃⁻ concentrations and isotopes over the GP16 cruise track (panel (a) inset) in the South Pacific. Profiles are presented from east (right) to west (left). Comparison of (a) observed [NO₃⁻] to (b) modeled [NO₃⁻] is presented over the full depth range (0-6000m). Comparison of (c) observed δ¹⁵N$_{NO3-}$ to (d) modeled δ¹⁵N$_{NO3-}$ is presented over a shortened depth range (0-1000m) to better assess surface and ODZ values. GEOTRACES data are from Peters et al. (2018a) and available from BCO-DMO.



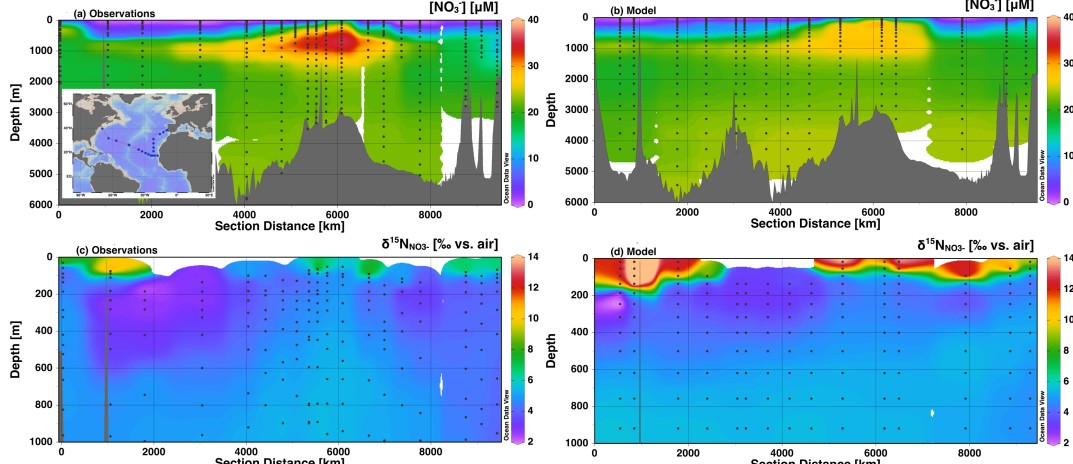

**Figure 6: Section profiles of NO₃⁻ concentrations and isotopes over the GA03 cruise track (panel (a) inset) in the North Atlantic. Profiles are presented from east (right) to west (left) from 0-6000km section distance, and then from south to north. Comparison of (a) observed [NO₃⁻] to (b) modeled [NO₃⁻] is presented over the full depth range (0-6000m). Comparison of (c) observed δ¹⁵N_NO₃⁻ to (d) modeled δ¹⁵N_NO₃⁻ is presented over a shortened depth range (0-1000m) to better assess surface and the low δ¹⁵N_NO₃⁻ contribution from N₂ fixation. GEOTRACES data are from Marconi et al. (2015) and available from BCO-DMO.**





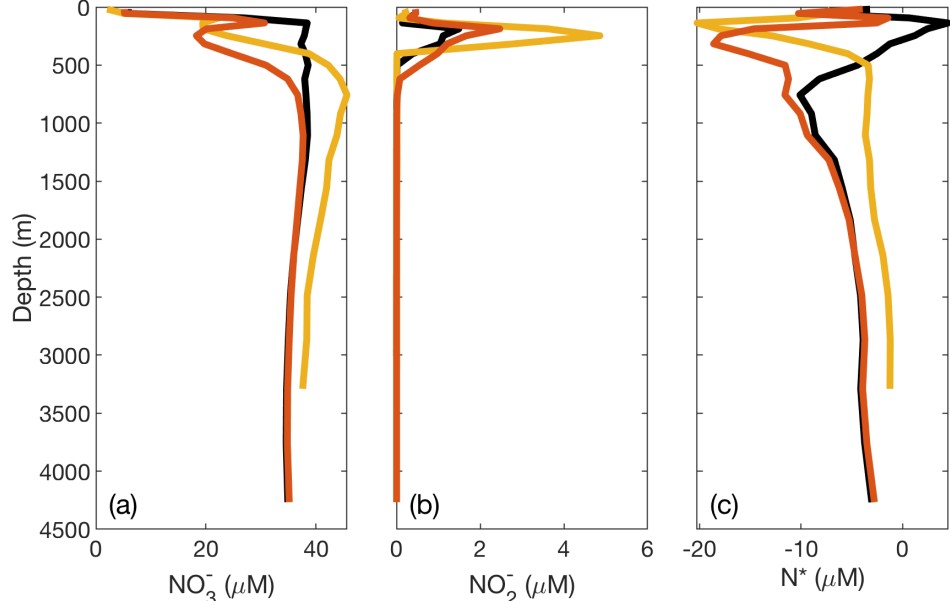

**Figure 7: Plot of DIN concentrations and N\* from the ETSP ODZ comparing modified O₂ thresholds for N loss. In the original optimized version of the model, there is insufficient N loss and NO₂⁻ accumulation in the ETSP. To demonstrate that this issue may be caused by issues with the gridded averages of O₂ and model grid size in the ETSP, we raised the O₂ thresholds for N loss-related processes (NAR, NIR, and AMX) to 15 μM. This effectively lowers the observed [O₂] in order to stimulate N loss. The resulting (a) [NO₃⁻], (b) [NO₂⁻], and (c) N\* are shown with the observed values from the database (yellow), original optimized model values (black), and lowered O₂ threshold model values (red).**





| Parameter | Value | Reference |
|---|---|---|
| b | -0.858 | Martin et al., 1987 |
| $F_0$ | 1.5 mmol N m$^{-3}$ yr$^{-1}$ | DeVries et al., 2013<br>Capone et al., 2005 |
| λ | 10 mmol N m$^{-3}$ | Holl & Montoya, 2005 |
| $T_0$ | 20 ˚C | DeVries et al., 2013<br>Capone et al., 2005 |
| $K_{Fe}$ | $4.4 \times 10^{-5}$ mmol Fe m$^{-3}$ | Follows et al., 2007 |
| $K_P$ | 0.005 mmol PO$_4^{3-}$ m$^{-3}$ | Moore & Doney, 2007 |
| $r_{C:N}$ | 6.625 | Redfield et al., 1963 |
| $K_m^{AMO}$ | 3.5 μM O$_2$ | Peng et al., 2016 |
| $K_m^{NXR}$ | 0.8 μM O$_2$ | Bristow et al., 2016b |
| $O_2^{NAR}$ | 7 μM O$_2$<br>15 μM O$_2$[a] | Dalsgaard et al., 2014; Jensen et al., 2008;<br>Kuypers et al., 2005; Kalvelage et al., 2011 |
| $O_2^{NIR}$ | 5 μM O$_2$<br>15 μM O$_2$[a] | Bonin et al., 1989; Kalvelage et al., 2011 |
| $O_2^{AMX}$ | 10 μM O$_2$<br>15 μM O$_2$[a] | Dalsgaard et al., 2014; Jensen et al., 2008;<br>Kuypers et al., 2005; Kalvelage et al., 2011 |
| $\delta^{15}N_{dep}$ | -4‰ | Hastings et al., 2009 |
| $\delta^{15}N_{fix}$ | -1‰ | Hoering & Ford, 1960; Carpenter et al., 1997 |
| $\alpha_{AMX,NO2}$ | 1.016 | Brunner et al., 2013 |
| $\alpha_{AMX,NXR}$ | 0.969 | Brunner et al., 2013 |
| $\alpha_{AMX,NH4}$ | 1 | |
| $\alpha_{AMO}$ | 1 | |
| $\alpha_{sed}$ | 1 | Brandes & Devol, 1997; Lehmann et al., 2004 |

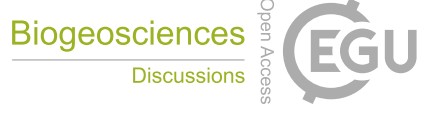

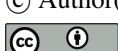

| | | |
|---|---|---|
| $\alpha_{assim}$ | 1.004 | Granger et al., 2010 |
| $\alpha_{remin}$ | 1 | Casciotti et al., 2008; Möbius, 2013 |
| $\alpha_{sol}$ | 1 | Knapp et al., 2011 |

    a.   Value used in test of ETSP N loss issue (Section 4.2)

**Table 1. Non-optimized model parameters.**





| Parameter | Initial | Final (avg.) | Error (2σ) | Final (global best fit) |
|---|---|---|---|---|
| $^{14}k_{PON}$ (yr$^{-1}$) | 3.9 | 3.9 | 0 | 3.9 |
| $^{14}k_{DON}$ (yr$^{-1}$) | 1.8 | 0.8 | 0.2 | 0.6 |
| $^{14}k_{NXR}$ (yr$^{-1}$) | 6.0 | 16.0 | 3.0 | 18.7 |
| $^{14}k_{NAR}$ (μM$^{-1}$ DON) | 2.5 | 1.6 | 0.8 | 2.3 |
| $^{14}k_{NIR}$ (μM$^{-1}$ DON) | 1.5 | 1.7 | 1.0 | 2.6 |

**Table 2. Optimized model parameters.**


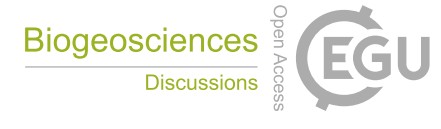

| Parameter | Values | References |
|-----------|--------|------------|
| $\varepsilon_{NAR}$ | 13‰, 25‰ | Granger et al., 2008<br>Kritee et al., 2012<br>Casciotti et al., 2013<br>Marconi et al., 2017 |
| $\varepsilon_{NIR}$ | 0‰, 15‰ | Casciotti et al., 2013<br>Martin and Casciotti, 2016 |
| $\varepsilon_{NXR}$ | -32‰, -20‰, -13‰ | Casciotti, 2009<br>Buchwald and Casciotti, 2010<br>Casciotti et al., 2013 |

**Table 3: Isotope effect cases.**