# Peer review of "Modeling oceanic nitrate and nitrite concentrations and isotopes using a 3D inverse N cycle model"

_Biogeosciences, 2018_

## Referee Comment (RC1) · I. Ruvalcaba Baroni (Referee) · 23 Oct 2018

Nitrogen plays a critical role in regulating oceanic primary productivity and there is an urgent need to improve our modeling technics to better predict future changes in the nitrogen cycle, in particular with expanding oxygen minimum zones. This study examines the importance of including nitrite in biogeochemical models in order to improve our understanding of the marine N cycle as well as to better capture nitrogen trends in the ocean. The explicit simulation of nitrite and its isotopic composition is a novel concept in this study, which was successfully applied.

Overall, the study is excellent. The model parametrization, the results and interpretations are all reasonable, and the manuscript is well-written and well-illustrated. I

recommend publication following minor revision, which mainly consist of rephrasing, clarifications and few suggestions.

Itzel Ruvalcaba Baroni

Stockholm University

**General comments**

While the manuscript is well-written, it is rather complex and heavy to read. I highly recommend reducing the number of acronyms used in the manuscript. This will greatly ease the reading and help its understanding.

I would recommend adding a short sentence on why it is important to improve the N cycle in models in the abstract (and not only in the introduction).

**Specific comments**

page 2:
line 16: Rephrase: e.g., N losses occur only where NO2 oxidation is limited by oxygen.

line 17: change to "critical to asses" ...

page 3:
line 24: change to: "inversion without the need for a spin-up period as required by forward models".

line 31: At this point, it is not entirely clear how the differences are optimized. The fact that it was done through sensitivity analysis comes too late in the paragraph. Maybe it should be mentioned earlier (beginning of the paragraph) that the optimization is done through sensitivity analysis.

Page 4:
line 4: To improve the reading is best to put it into words: Eg.g., Parameters that varied more than 5%, were...

line 30: It should be mentioned that ammonium can be important in some OMZ (see for example Bristow et al., 2016)

Page 6:
line 15: value of 0.003676 (equation 6). The reference for this value is missing.

page 7:
line 16: Is there a better symbol for pe? Another symbol would make it clear that pe is a value and not a word or typo.

line 22: avoid start a sentence with a symbol. Add a comma after equation 8 and "where": "0.419, where $\phi$ has a constant value of ..."

section 2.3.2:

What about riverine input of NO3, NO2 and PON? If not accounted in the model, please add a sentence of why is this not relevant for this study.

page 8:
line 4: Why must the assimilation rates for DON and PON be calculated using observed surface [NO3-]?

page 9:
line 1: How is this a refinement? Do you mean an improvement relative to what other model assume? In such case, how does a constant 14ksol improves PON solubilization?

page 11:
line 1-3: This should come before all oxygen dependent processes and maybe even be mentioned in section 2.1.

Page 12:
line 3: Please rephrase or explain what is meant by "cutoff points"

Page 13:

line 21: Which 200, 160 boxes? This si not clear, as these specific numbers have not been addressed before.

Page 17:
line 27: Replace lessened by "low".

lines 21-31: Could this also be due to the fact that ammonium, and therefore, ammonification and nitrification are not explicitly included in your model? Nitrification and ammonium uptake, have large fractionation effect. To which extent could this be important, in particular for low oxygen zones? (See e.g., Hoch et al., 1992, Higgins et al., 2011, Bristow et al., 2016 and Ruvalcaba Baroni et al., 2015)

page 19:
line 25: It is not clear in this sentence why slow rates will lead to more NO2

Page 20:
line 14: Use full names instead of acronyms, especially in the conclusions

Figure 4: It would be easier for the reader to have the colour legend as well directly in the figure.

---

## Referee Comment (RC2) · A. Bourbonnais (Referee) · 2 Nov 2018

**General comments**

In their paper, Martin *et al.* present a new global inverse model that represents the distribution of nitrate and nitrite concentrations and isotopes ($^{14}$N and $^{15}$N) in the ocean. The model is novel as it accounts for $NO_2^-$, a key intermediate in the marine N cycle, which is usually not considered. Overall, the manuscript was well written, easy to follow and the model and results were well discussed. This model will undoubtedly improve our understanding of the global marine N cycling, especially in ODZs. I strongly recommend its publication, after the relatively minor revisions below.

**Specific comments**
**Introduction**

**Page 3, lines 2-5:** How could these differences be reconciled? Could it be that the isotope effect for nitrite oxidation is determined by environment conditions and/or microbial assemblages? The kinetic isotope effect for nitrite oxidation needed in the model is more consistent with the large inverse ε for $NO_2^-$ oxidation of $\sim -30$ ‰ that has been observed for anammox bacteria in culture (Brunner *et al.*, 2013). Other processes, for instance nitrite oxidation with alternate electron acceptors, could also occur with a distinct (and as yet uncharacterized) isotope effect (e.g., Babbin *et al.*, 2017).

**Methods**
**Page 4, line 20:** I presume that the resolution of this model (2º x 2º) is currently too coarse to capture N-cycling in coastal regions where there is a high spatial (and temporal) heterogeneity, as well as higher turnover rates for most N-species (e.g. Hu *et al.*, 2016). Would it be feasible to increase the resolution in future versions of the model to better represent coastal regions?

**Page 4, lines 30-32:** $NH_4^+$ significantly accumulates in coastal regions, for instance, concentrations of up to 4 μM were observed off Peru in December 2012 (Hu *et al.*, 2016).

**Page 5, line 23:** Define WOA (World Ocean Atlas) when it is first used.

**Pages 5 lines 28-30 and page 6, lines 1-2:** Are modeled DON concentrations in surface waters consistent with a mostly recalcitrant DON pool in the ocean? The fast cycling for the labile fraction prevents its accumulation (Knapp *et al.*, 2005; Bourbonnais *et al.*, 2009). What is their modeled DON distribution?

**Page 6, line 9:** Are there any newer data for atmospheric N deposition? How well do these modeled estimates match real observations?

**Page 6, lines 25-28 and Page 7, lines 1-2:** Other organisms (not only *Trichodesmium*) mediate $N_2$ fixation. For instance, the distribution of unicellular cyanobacteria contrast with that of *Trichodesmium*, and depend on depth, temperature and water column density structure (Moisander *et al.*, 2010).

**Page 7, lines 12-13:** Is $N_2$ fixation in their model consistent with direct measurements? For instance, while previous models suggested significant $N_2$ fixation in the ETSP (Deutsch *et al.*, 2001), Knapp *et al.*, 2016, concluded that $N_2$ fixation is negligible there based on evidence from direct field measurements.

**Page 8, lines 6-7:** Is this a reasonable assumption? $NO_2^-$ accumulates significantly between 100 and 200 m depth in the ETSP ODZ (up to about 12 $\mu M$) (Bourbonnais *et al.*, 2015).

**Page 8, lines 30-31:** The Martin *et al.* (1997) mineralization curve requires a correction under anoxic condition where observed carbon burial is higher (e.g., Katsev and Crowe, 2015).

**Page 10, lines 14-15:** What about chemolithoautotrophic denitrification in sulfidic environments (e.g., Lavik *et al.*, 2009)?

**Page 11, line 1:** These thresholds are on the high side compared to the nanomolar ranges suggested by Dalsgaard *et al.*, 2014.

**Page 11, line 13:** "Anammox also produces 0.3 moles of $NO_3^-$…" Is this always true or it depends on environmental conditions?

**Page 12, lines 10-13:** Nitrification-denitrification in the sediments produces an efflux of heavier $\delta^{15}N-NH_4^+$ that would increase $\delta^{15}N-NO_3^-$ following nitrification (e.g., see Granger *et al.,* 2011).

**Page 14, lines 9-10:** It is preferable avoiding citing a manuscript in preparation.

**Results**
**Page 15, lines 22-25:** Are these relationships significant? It does not seem to be the case for nitrite (Fig. 3, b).

**Page 16, lines 17-18:** The model also fails to account for the episodic upwelling events in the ETSP and their effects on primary productivity and N-cycling.

**Page 16, lines 20-23:** N* is also affected by sedimentary processes in anoxic sediments, i.e., preferential release of $PO_4^{3-}$ due to oxyhydroxide dissolution (Noffke *et al.,* 2012).

**Page 17, line 2:** The model does not reproduce the extreme $\delta^{15}N-NO_3^-$ values (up to about 70‰ observed in Bourbonnais *et al.*, 2015) in the ETSP near the center of an anticyclonic eddy.

**Discussion**
**Page 19, lines 2-3:** Is this a reasonable rate?

**Page 19, lines 8-21:** Another consideration is that the $[O_2]$ data from the WOA were generally measured using Seabird sensors, with typical detection limit in the $\mu M$ range, while N-loss processes are inhibited at $[O_2]$ in the nM range.

**Conclusion**
**Page 20, lines 12-13:** Bourbonnais *et al.* (2015) also calculated an isotope effect of 16‰ for $NO_3^-$ reduction assuming a closed system.

**Figure 3:** Add $r^2$ values and degree of significance.

**Figure 4:** I assume that these are offshore profiles?

**Figures 5 and 6:** What about $NO_2^-$ and $\delta^{15}N\text{-}NO_2^-$?

**Technical corrections:**

**Page 9, line 15:** $NH_3$ appears to be in a different font.

**Page 21, line 8:** Remove the first "be" in "… given process may  vary, or be expressed …"

**Additional references**

Babbin, A. R., Peters, B. D., Mordy, C. W., Widner, B., Casciotti, K. L., & Ward, B. B. (2017). Multiple metabolisms constrain the anaerobic nitrite budget in the Eastern Tropical South Pacific. *Global Biogeochemical Cycles*, *31*(2), 258-271.

Bourbonnais, A., Lehmann, M. F., Waniek, J. J., & Schulz-Bull, D. E. (2009). Nitrate isotope anomalies reflect $N_2$ fixation in the Azores Front region (subtropical NE Atlantic). *Journal of Geophysical Research: Oceans*, *114*(C3).

Deutsch, C., Gruber, N., Key, R. M., Sarmiento, J. L., & Ganachaud, A. (2001). Denitrification and $N_2$ fixation in the Pacific Ocean. *Global Biogeochemical Cycles*, *15*(2), 483-506.

Granger, J., Prokopenko, M. G., Sigman, D. M., Mordy, C. W., Morse, Z. M., Morales, L. V., … & Plessen, B. (2011). Coupled nitrification-denitrification in sediment of the eastern Bering Sea shelf leads to $^{15}N$ enrichment of fixed N in shelf waters. *Journal of Geophysical Research*: Oceans, 116(C11).

Katsev, S., & Crowe, S. A. (2015). Organic carbon burial efficiencies in sediments: The power law of mineralization revisited. *Geology*, *43*(7), 607-610.

Knapp, A. N., Sigman, D. M., & Lipschultz, F. (2005). N isotopic composition of dissolved organic nitrogen and nitrate at the Bermuda Atlantic Time-series Study site. *Global Biogeochemical Cycles*, *19*(1).

Knapp, A. N., Casciotti, K. L., Berelson, W. M., Prokopenko, M. G., & Capone, D. G. (2016). Low rates of nitrogen fixation in eastern tropical South Pacific surface waters. *Proceedings of the National Academy of Sciences*, *113*(16), 4398-4403.

Lavik, G., Stührmann, T., Brüchert, V., Van der Plas, A., Mohrholz, V., Lam, P., ... & Kuypers, M. M. (2009). Detoxification of sulphidic African shelf waters by blooming chemolithotrophs. *Nature*, *457*(7229), 581.

Moisander, P. H., Beinart, R. A., Hewson, I., White, A. E., Johnson, K. S., Carlson, C. A., ... & Zehr, J. P. (2010). Unicellular cyanobacterial distributions broaden the oceanic $N_2$ fixation domain. *Science*, *327*(5972), 1512-1514.

Noffke, A., Hensen, C., Sommer, S., Scholz, F., Bohlen, L., Mosch, T., Graco, M., and Wallman, K.: Benthic iron and phosphorus fluxes across the Peruvian oxygen minimum zone, *Limnol. Oceanogr.*, 57, 851–867, 2012.

---

## Author Comment (AC2) · 11 Dec 2018

Page 11, Lines 1-3: The response to this comment was not addressed in the previous response. A mention of oxygen as a field that is not explicitly modeled has been added to the end of Section 2.1 (the N cycle model overview).

---

## Author Response (AR1)

**File contents**

- Response to Reviewer 1
- Response to Reviewer 2
- Brief list of changes in manuscript and supplement
5     • Marked up edited manuscript
- Marked up edited supplement

**Response to Reviewer 1**

Balancing the number of acronyms used in a manuscript is always challenging. In response to this comment, we have reduced the number of acronyms in the conclusions section. We find that using the acronyms for each process (e.g. NAR and NIR) in the rest of the text is less confusing than spelling out $NO_3^-$ reduction and $NO_2^-$ reduction, since it is easier to mistake the 3 and 2 in the subscripts than the "A" and "I" in the acronyms. There is no perfect solution to this, and we hope that removing the process acronyms in the conclusions will be helpful in this regard. In addition, we have opted to retain widely used three-letter acronyms for chemical species (DIN, DON, PON, etc.), but have revised the less common two-letter acronym 'ON' to 'organic N' throughout. We believe this strikes a balance between both clarity and brevity.

A brief comment on the importance of improving N cycle models by adding $NO_2^-$ as a tracer has been added to the abstract.

**Specific comments**

- Page 2, line 16: Rephrased as requested.
- Page 2, line 17: Rephrased as requested.
- Page 3, line 24: Rephrased as requested.
- Page 3, line 31: The optimization itself was not conducted using the sensitivity analysis, but the optimization procedure is discussed in Section 2.6. Sensitivity analysis was performed in order to narrow the set of parameters that was optimized, since there are too many parameters in the model to optimize them all concurrently due to computational limits.
- Page 4, line 4: Rephrased as requested.
- Page 4, line 30: Additional comments added to reference $NH_4^+$ accumulation in ODZs.
- Added text: "Though $NH_4^+$ has been observed to accumulate to micromolar concentrations in ODZs (Bristow et al., 2016; Hu et al., 2016), this occurs largely in shallow, coastal shelf regions, which are not resolved by the model."
- Page 6, line 15: References added for $r_{air}$ (Mariotti, 1983).
- Page 7, line 16: The references to the pe ratio have been changed to $P_e$ in both equations and the text to improve clarity.
- Page 7, line 22: Sentence has been restructured to avoid starting with a symbol
- Section 2.3.2: A brief discussion of the exclusion of riverine N inputs from the model has been added to the beginning of the section.
  - Added text: "In our model, atmospheric deposition and $N_2$ fixation are the only sources of bioavailable N in the model. These are the two largest sources of N to the ocean (Gruber and Galloway, 2008). We do not consider the third largest source of N, riverine fluxes, in the model due to lack of coastal resolution and the expectation that much of the river-derived N is denitrified in the shelf sediments (Nixon et al., 1996;

Seitzinger and Giblin, 1996). This would potentially impact the surface $NO_3^-$ and $\delta^{15}N_{NO3}$ but the overall contribution to the N budget would be negligible, especially considering the coarse resolution of the model."

- Page 8, line 4: DON and $NO_3^-/NO_2^-$ must be considered separately in order to introduce dependence on two N species for $NO_3^-$ reduction and $NO_2^-$ reduction, since these heterotrophic processes require both organic N and either $NO_3^-$ or $NO_2^-$. In order to use the simple linear model setup, DON and $NO_3^-/NO_2^-$ cannot be variables/unknown at the same time. Additional explanation can be found in Section 2.3.1, and some additional discussion has been added to the line in question.
  - Added text: "The assimilation rates for DON and PON must be calculated using observed surface $[NO_3^-]$, rather than modeled $[NO_3^-]$, in order for the heterotrophic processes of NAR and NIR to be dependent on both organic N and $NO_3^-$ or $NO_2^-$ availability, respectively."
- Page 9, line 1: This sentence has been clarified to indicate that having a variable $^{14}k_{sol}$ that accurately represents slower ON remineralization under low $O_2$ conditions would be a refinement that could be incorporated in future model versions.
  - Added text: "A spatially variable $^{14}k_{sol}$ that accounts for lower apparent values in ODZs is a refinement that could be introduced in future model versions."
- Page 11, line 1-3: A mention of oxygen as a field that is not explicitly modeled has been added to the end of Section 2.1 (the N cycle model overview).
- Page 12, line 3: Cutoff points discussion has been rephrased to clarify that they are the transition points between the piecewise segments of the $([O_2] - [NO_3^-])$ vs. sedimentary denitrification rate relationship.
  - Added text: "In order for sedimentary denitrification to be properly implemented in our linear model, we broke the original non-linear relationship into three roughly linear segments to create a piecewise relationship between $([O_2] - [NO_3^-])$ and sedimentary denitrification rate. We obtained three linear relationships between $([O_2] - [NO_3^-])$ and sedimentary denitrification rate, each applicable across a given range of $([O_2] - [NO_3^-])$ values (Figure S1). Due to the nature of our linear model, we needed to express the interval cutoff points that define the transition between the piecewise relationship segments in terms of $O_2$ rather than $([O_2] - [NO_3^-])$."
- Page 13, line 21: Sentence has been rephrased to indicate that there are 200,160 total ocean boxes in the model.
  - Added text: "All model ocean boxes (200,160 in total) are accounted for in the matrices."
- Page 17, line 27: Rephrased as requested.
- Page 17, lines 21-31: This section of text refers to model nitrate $\delta^{15}N$ that is lower than observations on a transect extending westward from the ETSP oxygen deficient zone, which we propose to be due to an underestimate of $NO_3^-$ reduction in the model ETSP. We interpret the reviewer's question to be about the potential role of fractionation during $NH_4^+$ production and consumption in driving $NO_3^-$ isotope variations in the real ocean, which are not fully

represented in the model. We agree that there typically are large fractionation factors associated with usage of $NH_4^+$ via uptake or oxidation, and that these fractionation factors should be expressed if $NH_4^+$ accumulates in the water column, or if there is a large difference in fractionation factors between competing pathways. The reviewer rightly points out that $NH_4^+$ is not explicitly included in the model. We feel justified in doing this since accumulation of $NH_4^+$ in the modern ocean, even in oxygen deficient zones, is typically very low. We agree that ignoring $NH_4^+$ in euxinic conditions, where it constitutes a large fraction of dissolved inorganic N, would not be advisable. In the current model, while not explicitly representing $NH_4^+$ as a DIN species, we do attempt to represent the partitioning of $NH_4^+$ generated during DON degradation (ammonification) between ammonia oxidation and anammox below the euphotic zone. From an isotope balance perspective, we also assume that these competing fates have similar fractionation factors, which, while fairly uncertain, is generally supported by available data. Here we apply similarly *low* fractionation factors (a = 1), which is not likely the case, but turns out to not impact nitrate $\delta^{15}N$ in the circumstances applied here, where $NH_4^+$ does not accumulate, and both consumption processes have the same isotope effect. As applied here, the $NO_3^-$ generated has the same $\delta^{15}N$ as remineralized DON. If an isotope effect were applied to ammonia oxidation, this would most likely lower the $\delta^{15}N$ of nitrate, not raise it. We believe that if anything our model results are biased towards he upper limit of nitrate $\delta^{15}N$ produced during nitrification. One scenario of concern for our application would be if anammox has a much larger fractionation factor towards $NH_4^+$ than ammonia oxidation, leaving $^{15}N$-enriched $NH_4^+$ to be oxidized to $NO_3^-$ in the water column. While theoretically possible, there are no observations that we are aware of that would support this scenario in the modern ocean. Secondly, if partly consumed $NH_4^+$ is released from the sediments and oxidized to $NO_3^-$ in the water column this could contribute to $^{15}N$-enrichment of $NO_3^-$ that is not represented by the model. There is observational support for this effect in the modern ocean, as mentioned by reviewer #2. In response to these two questions about the underrepresentation of $NH_4^+$ in the model, we have added a few additional comments regarding potential errors in the model associated with our representation.

- Added text:
  - p. 13: "This is a conservative estimate of the effects of benthic N loss on water column $NO_3^-$ isotopes, as several studies suggest that benthic N processes may contribute to water column nitrate $^{15}N$-enrichment (Lehmann et al., 2007; Granger et al., 2011; Somes et al., 2015; Brown et al., 2015). However, our current model parameterization does not require additional benthic fractionation to fit deep ocean $\delta^{15}N_{NO3}$. Also, our spatial resolution does not well represent regions where this effect might be important."
  - p. 18: "The simplification of $NH_4^+$ dynamics in the model could contribute to underestimation of $\delta^{15}N_{NO3}$ values if there were a large flux of $^{15}N$-enriched $NH_4^+$ from sediments (Granger et al., 2011), or if $^{15}N$-depleted $NH_4^+$ was preferentially transferred to the $N_2$ pool via anammox. While the isotope effect on $NH_4^+$ during anammox (Brunner et al., 2013) is higher than that applied here, we chose to balance this with a low isotope effect during aerobic $NH_4^+$ oxidation (Table 1)."

- Page 19, line 25: These sentences have been reworked to indicate that low $NO_2^-$ consumption rates (via NIR, AMX, and NXR) in conjunction with modest $NO_2^-$ production rates (via NAR and likely AMO) result in an accumulation of $NO_2^-$.
  - Added text: "The accumulation of $NO_2^-$ here in the model is likely due to $O_2$ concentrations falling below the set threshold for NAR but above the threshold for NIR, so $NO_2^-$ can accumulate via NAR but cannot be consumed via NIR. Although AMX and NXR occur there, the modeled rates of their $NO_2^-$ consumption are rather low, which in combination with high rates of NAR and no NIR leads to more $NO_2^-$ being produced than consumed."
- Page 20, line 14: Acronyms have been replaced with full process names in the conclusion.
- Figure 4: The figure has been amended to include a color legend for clarity (see attached image).

**Response to Reviewer 2**

- Page 3, lines 2-5: It is likely that the isotope effects measurements are subject to both environmental conditions and the microbial community composition. Though the strains chosen for culture by Buchwald and Casciotti (2010) and Casciotti (2009) are thought to be representative of the marine nitrite oxidizing community, that is not necessarily the case, and the pure culture studies in the laboratory are missing potential feedbacks from other *in situ* microbial processes or changing environmental conditions on an extremely local scale. There has been evidence for changing isotope effect expression under different environmental conditions (e.g. Krittee et al., 2012), so that could play a role in the different estimates. It is also possible that nitrite oxidation could be occurring with an alternative electron acceptor, but would be difficult to incorporate into our model due to our lack of thorough understanding about what these alternative pathways might be.

- Page 4, line 20: It would be a considerable computational challenge to increase the resolution of the physical circulation model. The data assimilation process used to construct the data-constrained tracer-transport model used in the present study took more than 1 month on a dedicated computer. With an increase in the horizontal resolution to $\sim 1/2°$ (the minimum resolution needed to start resolving the coastal ocean) the computational cost per iteration would increase by a factor of 16 and a larger number of iterations would likely be needed for the optimization to converge. Furthermore, the direct matrix inversions which we use to solve the physical and biogeochemical models would no longer be possible. The memory requirements would be too high. As a result, the fast direct solvers would have to be replaced by slower iterative solvers. Thus a global inverse model that resolves the coastal oceans is not presently feasible. It should however be feasible to embed our biogeochemistry model into a regional physical circulation model that resolves the coastal circulation.

- Page 4, lines 30-32: Additional comments added to reference $NH_4^+$ accumulation in ODZs.
  - Added text: "Though $NH_4^+$ has been observed to accumulate to micromolar concentrations in ODZs (Bristow et al., 2016; Hu et al., 2016), this occurs largely in shallow, coastal shelf regions, which are not resolved by the model."

- Page 5, line 23: A definition for the World Ocean Atlas acronym has been added to its first usage.

- Page 5, lines 28-30 and page 6, lines 1-2: DON in the model is not differentiated between labile and recalcitrant pools. The concentrations at the surface are largely between 0-20 μM, with some higher concentrations in equatorial regions with high surface production. The DON concentration decays rapidly with depth, as its distribution is controlled by PON solubilization (driven by a Martin curve) and remineralization with a first order rate constant. Though the concentration in the surface box is higher than it should be, averaging the concentration over the top two or three boxes (representative of the mixed layer) yields concentrations closer to 10 μM. The representation of ON in the model and the surface processes that affect it is relatively simplified, and could be improved and expanded in subsequent versions of the model.

- Page 6, line 9: The modeled N deposition values here match observations fairly well (see Dentener et al., 2006) though there is some spatial variation in the goodness of the prediction. Many of the regions with lower prediction accuracy are also those with lower fluxes, so the errors do not lead to large under- or over-predictions for global N deposition. There are some newer estimates for N deposition (as well as future projections) that will be explored in a subsequent paper.

- Page 6, lines 25-28 and page 7, lines 1-2: Though our $N_2$ fixation parameterization is by no means perfect, we do not exclude non-*Trichodesmium* $N_2$ fixers. Explictly adding additional types of $N_2$ fixers would require different parameters for maximum $N_2$ fixation rate, temperature constraints, and Fe and $PO_4^{3-}$ limitation, which we do not have good literature values for and are limited computationally by the number of parameters we can optimize. Additional investigations using this model that focus more on $N_2$ fixation rates could certainly implement different classes of $N_2$ fixers, but the generally good spatial patterns of $N_2$ fixation in the model and appropriate global rates (see below) are adequate for the ODZ analyses of interest here.

- Page 7, lines 12-13: The spatial patterns of $N_2$ fixation produced by the model are very similar to the observationally-constrained estimates of Luo et al. (2014). The modeled rates of $N_2$ fixation are appropriately low in the ETSP. Another paper (submitted to Global Biogeochemical Cycles) focuses on assessment of the N cycle process rates and their comparison to both other models and observational data.

- Page 8, lines 6-7: We believe that this is a reasonable assumption within the coarse framework of our model. Assimilation is only represented in the top two boxes of the model, which extend from 0-36 m (box 1) and 36-73 m (box 2). These two boxes are shallower than the characteristic high accumulation of $NO_2^-$ in the oxygen deficient zone, and accumulation of $NO_2^-$ in the primary $NO_2^-$ maximum usually only reaches a maximum of 1-2 μM, which would be negligible compared to $NO_3^-$.

- Page 8, lines 30-31: Though carbon burial is higher under anoxic conditions, there have been other work that has shown preferential mineralization of N relative to C under anoxic conditions (e.g. Van Mooy et al., 2002; Roberts et al., 2012). Since variable C:N ratios are beyond the scope of this model, and in order to keep things simple, we assume that the preferential mineralization of N relative to C balances the slower mineralization rate under anoxic conditions.

- Page 10, lines 14-15: We have revised this section to note that our estimate of denitrification does not include chemoautotrophic denitrification, citing Lavik et al., 2009. As our model does not include cycling of reduced sulfur species, it is not possible for us to include chemoautotrophic denitrification at this time, leading to our estimates potentially underestimating overall rates of denitrification. However, since our model is optimized to fit observations of $NO_3^-$ and $NO_2^-$ concentration and isotopes, the overall rate of denitrification is constrained. If chemolithotrophic denitrification can be assumed to have a similar N isotope effect as heterotrophic denitrification (Frey et al., 2014), it may just be part of the mix of our signal.
  - Added text: "When NAR occurs chemoautotrophically, it would be dependent primarily on the presence of $NO_3^-$ and an electron donor, such as hydrogen sulfide (Lavik et al., 2009). Since we do not model the

production of reduced sulfur species in our model, our estimates of denitrification would not explicitly include the effects of this process. However, chemolithotrophic denitrification could be tacitly accounted for in the optimization process, since the rate constants that control the rates of NAR and NIR are optimized in order to best fit the observations, and the isotope effect for chemolithotrophic denitrification is thought to be similar to that of heterotrophic denitrification (Frey et al., 2014)."

- Page 11, line 1: If the thresholds were set to nanomolar levels as suggested by Dalsgaard et al. (2014), there would be virtually no NAR, NIR, or AMX occurring in the model at all. The coarse resolution of the model results in depth-averaged $[O_2]$ in many ODZ boxes that are above zero, even after applying the empirical correction put forward by Bianchi et al. (2012) to improve WOA $O_2$ data fit in ODZs. It is not possible in this coarse, time-independent scheme to model any truly anoxic microenvironments. As a result, the area of the model ocean which is truly anoxic is very small. The area in which NAR, NIR, and AMX have been measured, observed, inferred, and modeled in previous ODZ studies is much greater than the extent of the anoxic model boxes, and thus we have chosen to be more flexible in our $O_2$ thresholds for anoxic processes in order to best reflect current knowledge of what is occurring within ODZs. As discussed in Section 4.2, the model dependency on this input $O_2$ has its downsides and does not adequately represent all ODZs despite our accommodating thresholds. The $O_2$ thresholds and the effect of changing the thresholds on process rates is discussed further in another paper (submitted to Global Biogeochemical Cycles).

- Page 11, line 13: Previous studies on the biomechanics of anammox have included the nitrate oxidoreductase enzyme, as the oxidation of nitrite via this enzyme produced electrons that are needed for autotrophic $CO_2$ fixation under normal anammox conditions (de Almeida et al., 2011; Kartal and Keltjens, 2016). Culture studies have also measured the stoichiometric production of $NO_3^-$, presumably by this enzyme (Strous et al., 1999; Brunner et al., 2013).

- Page 12, lines 10-13: This is an excellent suggestion, which we appreciate. We have thought about how to address this in the paper, as unfortunately it is not possible in the scope of the current model to incorporate this process in the model. The quoted text below represents our revisions to the paper in accordance.
  o Added text:
    ▪ p. 13: "This is a conservative estimate of the effects of benthic N loss on water column $NO_3^-$ isotopes, as several studies suggest that benthic N processes may contribute to water column nitrate [15]N-enrichment (Lehmann et al., 2007; Granger et al., 2011; Somes et al., 2015; Brown et al., 2015). However, our current model parameterization does not require additional benthic fractionation to fit deep ocean $\delta^{15}N_{NO3}$. Additionally, our spatial resolution does not well represent regions where this effect might be significant on bottom water $\delta^{15}N_{NO3}$, such as the shallow shelves."
    ▪ p. 18: "The simplification of $NH_4^+$ dynamics in the model could contribute to underestimation of $\delta^{15}N_{NO3}$ values if there were a large flux of [15]N-enriched $NH_4^+$ from sediments (Granger et al., 2011), or if [15]N-depleted $NH_4^+$ was preferentially transferred to the $N_2$ pool via anammox. While

the isotope effect on $NH_4^+$ during anammox (Brunner et al., 2013) is higher than that applied here, we chose to balance this with a low isotope effect during aerobic $NH_4^+$ oxidation (Table 1)."

- Page 14, lines 9-10: This paper has been submitted to Biogeosciences Discussions and will be available for proper citation shortly.
- Page 15, lines 22-25: The significance of these relationships was not calculated, and the lines shown are simply 1:1 lines to aid visual comparison between the modeled and observed data in the test set. It is likely that calculating such values would result in strange regression data and low significance. In many cases (particularly for $NO_2^-$), the magnitude of the model is incorrect, but observing $[NO_2^-]$ and $\delta^{15}N_{NO2}$ in a profile or section view reveals that the overall patterns are correct.
- Page 16, lines 17-18: The model cannot resolve episodic events, since it is a steady state, time-independent model that relies on a generalized circulation matrix. Though in the ETSP there is likely some upwelling that is incorporated into the circulation matrix, the more nuanced, time-dependent changes that occur in N cycling as a result of upwelling cannot be accounted for in this model.
  - Added text: "This could be due in part to the time-independent nature of this steady state inverse model, which does not capture the effects of upwelling events in the ETSP on N supply and cycling."
- Page 16, lines 20-23: An additional line accounting for the impact of sedimentary $PO_4^{3-}$ release on N* has been added, though attempting to account for said release is beyond the capabilities of this model. This is particularly true since the shallow shelves where low $O_2$ waters come into contact with sediments are not resolved in the model.
  - Added text: "Negative N* values are associated with N loss due to AMX or NIR or release of $PO_4^{3-}$ from anoxic sediments (Noffke et al., 2012), while positive N* values are associated with input of new N through $N_2$ fixation (Gruber and Sarmiento, 1997)."
- Page 17, line 2: It is not surprising that the model does not reproduce the high $\delta^{15}N_{NO3}$ values observed in the eddy for two reasons. First, the ETSP processes are not well-resolved by the model due to the aforementioned $O_2$ threshold issues. Second, the time-independent and steady-state nature of the model would not be able to capture transient features such as eddies, where the $\delta^{15}N_{NO3}$ increases over time due to lack of $NO_3^-$ resupply within the closed system of the eddy.
- Page 19, lines 2-3: This section has been rephrased. The original intent was to indicate the small fractional offset between the DIN and organic N model runs, both locally and globally.
  - Added text: "However, the majority of DIN assimilation estimates were within 10 μM/yr of the organic N production estimates, with an average offset of approximately 3.5% compared to DIN assimilation. The total global assimilation rates were within 0.4%, with some spatially variable differences due to offset between surface $[NO_3^-]$ and modeled $[NO_3^-]$. However we find that the WOA surface $NO_3^-$ values are fairly well represented by our modeled surface $NO_3^-$ (Figure S4)."

- Page 19, lines 8-21: The $O_2$ data product available through WOA is imperfect and coarse, which is why $O_2$ thresholds have been adjusted to accommodate the values. The empirical correction calculated by Bianchi et al. (2012) that corrects the WOA using the GLODAP data attempts to tackle this issue, but the $O_2$ values in ODZs are still too high. It is likely that if we had more thorough spatial coverage of $O_2$ data from STOX sensors (rather than Seabird), we could use the model to more accurately and thoroughly probe some of the questions about $O_2$ limitation for these processes. However, using a time-independent, steady-state model would still be unable to capture small, dynamic changes in $O_2$ that may be important for driving some N cycling within regions that are close to the $O_2$ detection limit or the $O_2$ limit for these processes.
- Page 20, lines 12-13: Additional reference to Bourbonnais et al. (2015) has been added.
- Figure 3: The lines shown are not regressions, they are 1:1 lines and do not have associated $r^2$ or significance values. See above for further discussion of the comparison between model output and test set data.
- Figure 4: These are indeed offshore profiles. Due to low model resolution, there are very few coastal or on-shelf boxes, especially in ODZs. A note has been added to the figure legend and description to indicate that the profiles are offshore.
- Figure 5-6: Section profiles comparing the modeled and observed $[NO_2^-]$ and $\delta^{15}N_{NO2}$ for the GP16 transect have been added to the supplement (and are attached to this comment). $NO_2^-$ does not accumulate along the GA03 transect and no $\delta^{15}N_{NO2}$ measurements were made, so those profiles are not included.
- Page 9, line 15: This font error and a few others were corrected.
- Page 21, line 8: This grammatical error was corrected.

**Brief list of changes in manuscript**

- Title has been updated to include nitrate
- Minor textual changes for clarity, including reducing the number of acronyms used
- Some discussion of $N_2$ fixation has been added, with an additional figure and table included in the supplement
5
- Supplemental figure with GP16 $NO_2^-$ concentration and isotopes has been added
- Discussion of the treatment of $NH_4^+$ has been added throughout the manuscript, as it is one of the major model caveats
- Figure 4 has been updated to include a legend
- References have been added to further justify model assumptions and address shortcomings with this model setup

[revised manuscript text omitted]

| Parameter | Values | References |
|---|---|---|
| $\epsilon_{NAR}$ | 13‰, 25‰ | Granger et al., 2008
Kritee et al., 2012
Casciotti et al., 2013
Marconi et al., 2017 |
| $\epsilon_{NIR}$ | 0‰, 15‰ | Casciotti et al., 2013
Martin and Casciotti, 2016 |
| $\epsilon_{NXR}$ | -32‰, -20‰, -13‰ | Casciotti, 2009
Buchwald and Casciotti, 2010
Casciotti et al., 2013 |

**Table 3: Isotope effect cases.**

**Section S1**

Linearization of transfer function for sedimentary denitrification

This section expands on the transformations applied to the non-linear transfer function for sedimentary denitrification presented by Bohlen et al. (2012) in order to use the transfer function in our linear N cycle model. The original function is as

5    follows:

1. DIN loss = $(0.60 + 0.19*0.99^{(O2-NO3)bottom})$*RRPOC    for water depths >= 1000m
2. DIN loss = $0.73*(0.60 + 0.19*0.99^{(O2-NO3)bottom})$*RRPOC        for water depths < 1000m

10    The calculation of the rain rate of particulate organic carbon (RRPOC) follows the Martin curve is as described in Text S4 (Equation 20). $(O_2\text{-}NO_3^-)_{bottom}$ is the difference between $[O_2]$ and $[NO_3^-]$ at the bottom of the water column, where it interfaces with the sediments.

For every model grid box, the depth is taken to be the depth at the bottom of the box. Each box is then assigned a multiplier of

15    1 (if depth >= 1000m) or 0.73 (if depth < 1000m) that will be multiplied by the sedimentary denitrification terms.

The next step is linearizing the (DIN loss)/RRPOC data presented by Bohlen et al. (2012) with respect to $(O_2\text{-}NO_3^-)_{bottom}$, since we cannot use the exponential equation in our linear system. This was performed by selecting two $(O_2\text{-}NO_3^-)_{bottom}$ cutoff points (29 μM and 141 μM), breaking the data into three groups. A piecewise linear regression was then performed on each of these

20    sections (Figure S1), resulting in the following equations:

3. $(DIN\ loss)/RRPOC = 0.297 - 0.005(O_2\text{-}NO_3^-)_{bottom}$    $(O_2\text{-}NO_3^-)_{bottom} <= 29$ μM
4. $(DIN\ loss)/RRPOC = 0.222 - 0.001(O_2\text{-}NO_3^-)_{bottom}$    29 μM $< (O_2\text{-}NO_3^-)_{bottom} <= 141$ μM
5. $(DIN\ loss)/RRPOC = 0.105 - 0.000006(O_2\text{-}NO_3^-)_{bottom}$    141 μM $< (O_2\text{-}NO_3^-)_{bottom}$

These $(O_2\text{-}NO_3^-)_{bottom}$ cutoff points were then converted to $O_2$ cutoff points in order to use a simple N-independent mask to determine which of the relationships to apply to a given model grid box. A linear relationship between $[O_2]$ and $(O_2\text{-}NO_3^-)_{bottom}$ was determined using The 2013 World Ocean Atlas interpolated data product for $[O_2]$ and $[NO_3^-]$ (Garcia et al., 2014). The linear relationship is as follows and is also shown in Figure S2:

6. $(O_2\text{-}NO_3^-)_{bottom} = 1.12[O_2]_{bottom} -55.6$

The $(O_2\text{-}NO_3^-)_{bottom}$ cutoff points can then be expressed in $[O_2]$ units as 75 and 175 μM.

The final step in modifying this transfer function for use in the linear model is to break the piecewise linear equations into a component that is dependent on N and a component that is independent of N. This facilitates the implementation of the equations in our linear system.

7. Independent + dependent = (DIN loss)/RRPOC

8. Independent = 0.297 – 0.005[$O_2$]          $O_2 <= 75$ μM
9. Dependent = 0.005[$NO_3^-$]

10. Independent = 0.222 – 0.001[$O_2$]          75 μM < $O_2$ <= 175 μM
11. Dependent = 0.001[$NO_3^-$]

12. Independent = 0.105 – 0.000006[$O_2$]          175 μM < $O_2$
13. Dependent = 0.000006[$NO_3^-$]

[Figure]

**Figure S1**

(a) Map of modeled areal $N_2$ fixation rates. (b) Map of the fraction of N input due to atmospheric deposition of DIN, with the remaining fraction due to $N_2$ fixation.

[Figure]

**Figure S2.** Piecewise division of the transfer function for sedimentary denitrification from Bohlen et al. (2012). In order to incorporate this into our liner model, we split the original non-linear relationship between sedimentary denitrification rate, rain rate of particulate organic carbon (RRPOC), and bottom water ($[O_2] - [NO_3^-]$) into three linear segments with cutoff points in terms of ($[O_2] - [NO_3^-]$). These cutoff points were then converted to $[O_2]$ cutoff points using a relation shown in Figure S2.

[Figure]

y = 1.12x + -55.6

**Figure S3.** Plot of 2013 World Ocean Atlas [O$_2$] vs. ([O$_2$] – [NO$_3^-$]) (Garcia et al., 2014). In order to express the sedimentary denitrification transfer function cutoff points (Figure S1) in terms of [O$_2$] rather than ([O$_2$] – [NO$_3^-$]), we determined a linear relationship between [O$_2$] and ([O$_2$] – [NO$_3^-$]).

[Figure]

**Figure S4.** Modeled (a) $[NO_3^-]$, (b) $[NO_2^-]$, (c) $\delta^{15}N_{NO3-}$, and (d) $\delta^{15}N_{NO2-}$ are compared against the corresponding values from the database training set. Shown on each panel is a 1:1 line starting at the origin. Data in black have corresponding $[O_2] < 10$ μM, and data in gray have $[O_2] \geq 10$ μM.

[Figure]

**Figure S5.** Map showing a comparison between modeled surface [NO₃⁻] for the top two model boxes and 2013 World Ocean Atlas [NO₃⁻] (Garcia et al., 2014) interpolated to the model grid for the same depths. Areas where the model overpredicts surface [NO₃⁻] are shown in yellow, and underprediction is shown in blue.

[Figure]

[Figure]

**Figure S6.** Section profiles of NO₂⁻ concentrations and isotopes over the GP16 cruise track (panel (a) inset) in the South Pacific. Profiles are presented from east (right) to west (left). Comparison of (a) observed [NO₂⁻] to (b) modeled [NO₂⁻] is presented over a shortened depth range (0-1000m) to better assess surface and ODZ values where NO₂⁻ accumulates. GEOTRACES data are from Peters et al. (2018a) and available from BCO-DMO.

**Figure S5.**

[Figure]

**Figure S7.** Difference in assimilation rates between the DIN and ON model runs plotted against the modeled [NO₃⁻]. Points are colored by the difference between modeled and observed [NO₃⁻].

[Figure]

**Figure S8.** Map of annual average 2013 World Ocean Atlas [$O_2$] @ 200m depth (Garcia et al., 2014) to demonstrate areas where $O_2$ is low enough for anoxic processes such as nitrate reduction, nitrite reduction, and anammox. The canonical ODZs are visible in blue: the Arabian Sea, ETNP, and ETSP. Also shown in blue are the Bay of Bengal and the Black Sea.

| Global | | |
|---|---|---|
| **N₂ fixation rate (Tg N/yr)** | **Method of estimation** | **Reference** |
| 137 | Ocean circulation model of P* | Deutsch et al., 2007 |
| 110 ± 40[a] | N* observations | Gruber & Sarmiento, 1997 |
| 74 (51-110) | Tracer incubations | Luo et al., 2014 |
| 131 | Global inverse N model | This study |
| **Pacific** | | |
| **N₂ fixation rate (Tg N/yr)** | **Method of estimation** | **Reference** |
| 95 | Ocean circulation model of P* | Deutsch et al., 2007 |
| 59 ± 14 | N* observations | Deutsch et al., 2001 |
| 37 (25-56) | Tracer incubations | Luo et al., 2014 |
| 67 | Global inverse N model | This study |
| **Atlantic** | | |
| **N₂ fixation rate (Tg N/yr)** | **Method of estimation** | **Reference** |
| 20 | Ocean circulation model of P* | Deutsch et al., 2007 |
| 30.5 ± 4.9 | Nitrate isotope mass balance | Marconi et al., 2017 |
| 27.6 ± 10 | Tracer incubations | Fonseca-Batista et al., 2017 |
| 28[a] | N* observations | Gruber & Sarmiento, 1997 |
| 13.6 (9.7-19.4) | Tracer incubations | Luo et al., 2014 |
| 32 | Global inverse N model | This study |

a.   Extrapolated from N. Atlantic estimate

b.   Only from 10˚N-50˚N

**Table S1.** Global N₂ fixation rate comparisons.

| Location and year sampled | Data types | Reference |
|---|---|---|
| ETNP, 2003 | $[NO_3^-]$, $[NO_2^-]$, $\delta^{15}N_{NO3-}$, $\delta^{15}N_{NO2-}$ | Casciotti and McIlvin, 2007 |
| ETNP, 2012 | $[NO_3^-]$, $[NO_2^-]$, $\delta^{15}N_{NO3-}$, $\delta^{15}N_{NO2-}$ | Casciotti, unpublished |
| ETSP, 2011 | $[NO_3^-]$, $[NO_2^-]$, $\delta^{15}N_{NO3-}$, $\delta^{15}N_{NO2-}$ | Casciotti, unpublished |
| ETSP, 2013 | $[NO_3^-]$, $[NO_2^-]$, $\delta^{15}N_{NO3-}$, $\delta^{15}N_{NO2-}$ | Peters et al., 2018 |

**Table S2.** New additions to the database originally compiled by Rafter et al. (in prep.).